# Triplets Better Than Pairs: Towards Stable and Effective Self-Play Fine-Tuning for LLMs

**Yibo Wang**[1,2,*], **Hai-Long Sun**[1,2,*], **Guangda Huzhang**, **Qing-Guo Chen**[3],
**Zhao Xu**[3], **Weihua Luo**[3], **Kaifu Zhang**[3], **Lijun Zhang**[1,4,2,†]

[1]National Key Laboratory for Novel Software Technology, Nanjing University, Nanjing, China
[2]School of Artificial Intelligence, Nanjing University, Nanjing, China
[3]Alibaba International Digital Commerce [4]Pazhou Laboratory (Huangpu), Guangzhou, China
{wangyb, sunhl, zhanglj}@lamda.nju.edu.cn
{qingguo.cqg, changgong.xz, weihua.luowh, kaifu.zkf}@alibaba-inc.com

## Abstract

Recently, self-play fine-tuning (SPIN) has been proposed to adapt large language models to downstream applications with scarce expert-annotated data, by iteratively generating synthetic responses from the model itself. However, SPIN is designed to optimize the current reward advantages of annotated responses over synthetic responses at hand, which may gradually vanish during iterations, leading to *unstable optimization*. Moreover, the utilization of reference policy induces a *misalignment* issue between the reward formulation for training and the metric for generation. To address these limitations, we propose a novel **T**riplet-based **S**elf-**P**lay f**I**ne-tu**N**ing (T-SPIN) method that integrates two key designs. First, beyond current advantages, T-SPIN additionally incorporates historical advantages between iteratively generated responses and proto-synthetic responses produced by the initial policy. Even if the current advantages diminish, historical advantages remain effective, stabilizing the overall optimization. Second, T-SPIN introduces the entropy constraint into the self-play framework, which is theoretically justified to support reference-free fine-tuning, eliminating the training-generation discrepancy. Empirical results on various tasks demonstrate not only the superior performance of T-SPIN over SPIN, but also its stable evolution during iterations. Remarkably, compared to supervised fine-tuning, T-SPIN achieves comparable or even better performance with only 25% samples, highlighting its effectiveness when faced with scarce annotated data.

## 1 Introduction

Large language models (LLMs) have demonstrated powerful generalization capabilities and profound impact across various fields [Ouyang et al., 2022, Huang and Chang, 2023, Qin et al., 2023, Bai et al., 2023, Achiam et al., 2023, Liu et al., 2023, Jiang et al., 2024a, Wu et al., 2024a, Zhao et al., 2024, Chen et al., 2024a, Liu et al., 2024, Hui et al., 2024, Guo et al., 2024a, Jiang et al., 2024b, Anthropic, 2024a, Google, 2024, OpenAI, 2024, Sun et al., 2025a,b, Lu et al., 2025, Jiang et al., 2025]. The remarkable success of LLMs is inherently linked to the utilization of vast expert annotations. Nevertheless, considerable curation costs for high-quality data pose major obstacles to downstream-task adaptation [Wang et al., 2024a, Tan et al., 2024]. Consequently, numerous efforts have been devoted to addressing this challenge, such as data augmentation [Dai et al., 2023, Zhu et al., 2023, Guo et al., 2024b] and model distillation [Gunasekar et al., 2023, Luo et al., 2024, Xin et al., 2024].

---

*Work done during the internship at Alibaba International Digital Commerce.
†Lijun Zhang is the corresponding author.

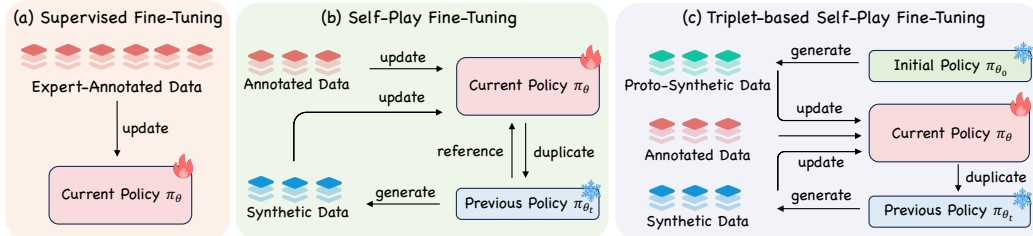

Figure 1: Comparisons of three strategies: (a) supervised fine-tuning requires large amounts of annotated data to train $\pi_\theta$; (b) self-play fine-tuning operates with limited annotated data and iteratively generated samples, and employs the previous policy $\pi_{\theta_t}$ as a reference for updates; (c) triplet-based self-play fine-tuning employs triplet inputs, i.e., annotated data, synthetic samples, and proto-synthetic ones from the initial policy $\pi_{\theta_0}$, and updates $\pi_\theta$ without auxiliaries from any reference policies.

Recently, a promising research direction known as self-play fine-tuning has been introduced [Chen et al., 2024b], and has since been applied to various LLM fields [Yuan et al., 2024a, Wu et al., 2024b, Gao et al., 2024a]. The core idea of self-play fine-tuning is to progressively refine the LLM by competing against itself. Formally, it can be framed as a two-player game between a main player and an opponent player, each of which corresponds to an instance policy of the same LLM. In this game, the main player aims to distinguish expert-annotated responses from generated ones, while the opponent player strives to produce high-quality synthetic responses that can deceive the main player.

Based on this framework, Chen et al. [2024b] introduce the seminal method named SPIN, which is designed to optimize the relative reward advantages of annotated responses over generated ones, and employs the policy from previous iteration as a reference for reward computations. However, with the evolution of LLM, the relative advantage between two types of data gradually diminishes over time. When the relative advantage is eliminated, the optimization objective of SPIN degenerates into a constant independent of the policy, allowing *any policy* to become an optimal solution and thus leading to *unstable optimization*. Moreover, the incorporation of reference policy in SPIN causes a *misalignment* between the reward formulation used for training and the log likelihood that guides response generation. Empirical evidences in Figure 3 reveal that although annotated responses yield higher rewards than generated responses, their log likelihoods become lower than synthetic samples during iterations. In other words, higher rewards do not necessarily lead to higher generation priority.

To address these limitations, we propose a new self-play fine-tuning method, termed **T**riplet-based **S**elf-**P**lay f**I**ne-tu**N**ing (T-SPIN). The basic intuition is to incorporate historical advantages for stable evolution, and eliminate the reference policy for training-generation alignments. Specifically, we introduce two key modifications to the self-play framework. First, the main player is required not only to distinguish between real and synthetic data, but also discriminate *proto-synthetic* samples produced by the initial policy. By incorporating triplet inputs of real, synthetic and proto-synthetic samples, T-SPIN optimizes current and historical advantages simultaneously. Even if the current advantage diminishes (i.e., synthetic samples become indistinguishable from annotated ones), the historical advantages between synthetic responses and proto-synthetic responses still exist, ensuring a stable optimization over the policy model. Second, the opponent player maximizes the confidence of synthetic responses with an *entropy constraint*, where the confidence measures how likely the main player believes a response comes from the annotated data. Through theoretical analysis, we demonstrate that the incorporation of the entropy constraint removes the need for reference policy, aligning the principles of training and generation. Empirical results show that T-SPIN consistently maintains higher log-likelihoods for annotated data compared to synthetic samples.

We conduct experiments on Zephyr-7B [Tunstall et al., 2023] and Mistral-7B [Jiang et al., 2023], and select $50k$ annotated samples from Ultrachat200k [Ding et al., 2023] as the training set. The performance is evaluated across diverse tasks to assess multi-dimensional capabilities of LLMs, including math and logic, multi-domain knowledge, commonsense reasoning, and instruction-following. Experimental results demonstrate that T-SPIN significantly improves performance, particularly in math and instruction-following, and achieves superior and stable performances during iterations compared to SPIN. Furthermore, we also show that T-SPIN, using only $25\%$ annotated samples, can achieve comparable performance (or even better on certain tasks) to supervised fine-tuning with the full training set, highlighting the effectiveness of T-SPIN in addressing the scarcity of annotated data.

## 2 Preliminary

In this section, we provide a brief overview of three fine-tuning strategies (i.e., supervised fine-tuning, self-play fine-tuning, and our T-SPIN) for LLMs. The comparisons are also illustrated in Figure 1.

### 2.1 Supervised fine-tuning

A common strategy to adapt a pretrained model to downstream tasks is fine-tuning it on high-quality annotated data. Specifically, given a prompt $\mathbf{x}$ and its corresponding annotated response $\mathbf{y}$, supervised fine-tuning (SFT) aims to align the model with $(\mathbf{x}, \mathbf{y})$. Formally, the goal of SFT is to learn a policy $\pi_{\text{SFT}}$ by maximizing the log-likelihood of high-quality annotated data:

$$\pi_{\text{SFT}} = \underset{\pi_\theta}{\arg\max}\, \mathbb{E}_{\mathbf{x} \sim q(\cdot), \mathbf{y} \sim \pi_{data}(\cdot|\mathbf{x})} \left[\log \pi_\theta(\mathbf{y}|\mathbf{x})\right], \tag{1}$$

where $q(\cdot)$ and $\pi_{data}(\cdot|\mathbf{x})$ denote the distributions of prompts and annotated responses, respectively. Mathematically, (1) is equivalent to the solution of minimizing the KL divergence between the target distribution $\pi_{data}(\cdot|\mathbf{x})$ and the policy $\pi_\theta(\cdot|\mathbf{x})$:

$$\pi_{\text{SFT}} = \underset{\pi_\theta}{\arg\min}\, \mathbb{E}_{\mathbf{x} \sim q(\cdot)} \left[\mathbb{D}_{\text{KL}}(\pi_{data}(\cdot|\mathbf{x})||\pi_\theta(\cdot|\mathbf{x}))\right]. \tag{2}$$

It can be verified that the solution (2) exactly matches the target distribution of annotated responses, i.e., $\pi_{\text{SFT}}(\cdot|\mathbf{x}) = \pi_{\text{data}}(\cdot|\mathbf{x})$. Unfortunately, achieving the optimal policy typically demands extensive high-quality samples, resulting in high costs for data curation and annotation, and thereby limiting the practical applicability of supervised fine-tuning [Wang et al., 2024a, Tan et al., 2024].

### 2.2 Self-play fine-tuning

Self-play fine-tuning has emerged as a promising way to address the scarcity of expert-annotated data when adapting LLMs to downstream tasks [Chen et al., 2024b, Yuan et al., 2024a]. Different from SFT that runs over a fixed annotated dataset, self-play fine-tuning employs both annotated samples and self-generated responses as inputs, with the latter continuously refined as the model evolves.

Specifically, at each iteration $t + 1$, the model first generates a synthetic response $\mathbf{y}'$ for a given prompt $\mathbf{x}$ by sampling from the policy $\pi_{\theta_t}(\cdot|\mathbf{x})$. Then, the model utilizes the prompt $\mathbf{x}$ and pairwise samples (i.e., the annotated response $\mathbf{y}$ and the synthetic response $\mathbf{y}'$) to obtain the policy $\pi_{\theta_{t+1}}$. Let the reward for a response be formulated by the log ratio of the likelihood between the current policy $\pi_\theta$ and the previous policy $\pi_{\theta_t}$ [Rafailov et al., 2023]. Then, the objective of SPIN [Chen et al., 2024b] is to maximize the gap (referred to as the relative advantage) between the reward of annotated response $\mathbf{y}$ and that of synthetic response $\mathbf{y}'$:

$$\pi_{\theta_{t+1}} = \underset{\pi_\theta}{\arg\min}\, \mathbb{E}\left[\ell\left(\lambda \log \frac{\pi_\theta(\mathbf{y}|\mathbf{x})}{\pi_{\theta_t}(\mathbf{y}|\mathbf{x})} - \lambda \log \frac{\pi_\theta(\mathbf{y}'|\mathbf{x})}{\pi_{\theta_t}(\mathbf{y}'|\mathbf{x})}\right)\right], \tag{3}$$

where $\lambda > 0$ denotes a hyper-parameter, $\ell(\cdot)$ is a convex and monotonically decreasing function, and the expectation operation is taken over the distributions $\mathbf{x} \sim q(\cdot), \mathbf{y} \sim \pi_{data}(\cdot|\mathbf{x})$ and $\mathbf{y}' \sim \pi_{\theta_t}(\cdot|\mathbf{x})$. In the following, we elaborate two major limitations of SPIN: *unstable optimization* during iterations and *misalignment* between training and generation.

**Unstable optimization.** The instability of SPIN arises from the inherent structure of the objective in (3). To be precise, we consider the case where the synthetic response $\mathbf{y}'$ becomes identical to the annotated response $\mathbf{y}$ (i.e., $\mathbf{y}' = \mathbf{y}$). In this case, the advantage of $\mathbf{y}$ over $\mathbf{y}'$ vanishes, causing the objective (3) to degenerate into a constant that is independent of the policy $\pi_\theta$. As a result, *any* policy $\pi_\theta$ becomes optimal to (3), leading to unstable optimization or even performance degradation.

**Misalignment between training and generation.** Given the reference policy $\pi_{\theta_t}$ from the iteration $t$, the objective (3) is designed to optimize the reward gap between two types of responses, where the reward for a response $\hat{\mathbf{y}}$ is defined as $r(\mathbf{x}, \hat{\mathbf{y}}) = \lambda \log \pi_\theta(\hat{\mathbf{y}}|\mathbf{x}) - \lambda \log \pi_{\theta_t}(\hat{\mathbf{y}}|\mathbf{x})$. However, it is important to note that the formulation of $r(\mathbf{x}, \hat{\mathbf{y}})$ differs from the principle used for generation, i.e., $\log \pi_\theta(\hat{\mathbf{y}}|\mathbf{x})$. In other words, due to the incorporation of reference policy $\pi_{\theta_t}$, a higher reward, i.e., $r(\mathbf{x}, \mathbf{y}) > r(\mathbf{x}, \mathbf{z})$ for two responses $\mathbf{y}$ and $\mathbf{z}$, does not necessarily imply a higher generation probability, i.e., $\log \pi_\theta(\mathbf{y}|\mathbf{x}) > \log \pi_\theta(\mathbf{z}|\mathbf{x})$ may not hold. This demonstration has been empirically verified by our experiments, as illustrated in Figure 3.

# 3 Triplet-based self-play fine-tuning

In this section, we introduce our triplet-based self-play fine-tuning (T-SPIN), a stable and reference-free method for fine-tuning LLMs. Building upon the self-play framework, our T-SPIN involves two players: a main player and an opponent player, both of which are different policy instances from the same LLM. For clarity, we describe the update rules for two players separately.

**The update rule for main player.** Recall that the goal of the main player is to distinguish among the annotated response $\mathbf{y}$, the synthetic response $\mathbf{y}'$ generated from the previous policy $\pi_{\theta_t}$ and the proto-synthetic response $\mathbf{y}_0$ produced by the initial policy $\pi_{\theta_0}$. Conceptually, the triple responses are expected to satisfy the relation $\mathbf{y} \succeq \mathbf{y}' \succeq \mathbf{y}_0$, where the operator $\succeq$ indicates that the main player has greater preference for one response than in another. In other words, $\mathbf{y}_0$ serves as an anchor that helps to evaluate how far the current policy $\pi_{\theta_t}$ has progressed. This design is partially inspired by the schema-equilibration process in cognitive development, where knowledge builds upon and refines pre-existing cognitive anchors [Piaget, 1952]. Formally, we are motivated by the integral probability metric (IPM) [Müller, 1997] to establish the objective of the main player. In detail, let $c(\mathbf{x}, \hat{\mathbf{y}})$ denote the confidence with which the main player believes a response $\hat{\mathbf{y}}$ is sampled from the real-world data distribution $\pi_{\text{data}}(\cdot|\mathbf{x})$. Then, the goal of main player is to find a confidence function

$$c_{t+1} = \underset{c \in \mathcal{C}}{\operatorname{argmin}} \, \mathbb{E}\left[\ell\left(c(\mathbf{x}, \mathbf{y}) - c(\mathbf{x}, \mathbf{y}')\right) + \beta \ell\left(c(\mathbf{x}, \mathbf{y}') - c(\mathbf{x}, \mathbf{y}_0)\right)\right], \tag{4}$$

where $\beta \geq 0$ denotes a trade-off hyper-parameter, and $\mathcal{C}$ denotes a function class that will be specified later, and $\ell(\cdot)$ denotes a monotonically decreasing function, e.g., the logistic function $\ell(x) = -\log \sigma(x)$ and the exponential function $\ell(x) = \exp(-x)$. The expectation operation here is taken over the distributions $\mathbf{x} \sim q(\cdot), \mathbf{y} \sim \pi_{data}(\cdot|\mathbf{x}), \mathbf{y}' \sim \pi_{\theta_t}(\cdot|\mathbf{x})$ and $\mathbf{y}_0 \sim \pi_{\theta_0}(\cdot|\mathbf{x})$. Given the triple responses $\mathbf{y}, \mathbf{y}'$, and $\mathbf{y}_0$, the objective in (4) consists of two terms:

- The first term $\ell\left(c(\mathbf{x}, \mathbf{y}) - c(\mathbf{x}, \mathbf{y}')\right)$ measures the gap between the real response $\mathbf{y} \sim \pi_{\text{data}}(\cdot|\mathbf{x})$ and the synthetic one $\mathbf{y}' \sim \pi_{\theta_t}(\cdot|\mathbf{x})$, reflecting the relative advantages of $\pi_{\text{data}}$ over $\pi_{\theta_t}$;
- The second term $\ell\left(c(\mathbf{x}, \mathbf{y}') - c(\mathbf{x}, \mathbf{y}_0)\right)$ compares the synthetic response $\mathbf{y}'$ to the proto-synthetic response $\mathbf{y}_0 \sim \pi_{\theta_0}(\cdot|\mathbf{x})$, capturing historical advantages of $\pi_{\theta_t}$ over $\pi_{\theta_0}$ during iterations.

During the self-play process, the opponent player evolves over time, generating synthetic responses $\mathbf{y}'$ that progressively approximate the real response $\mathbf{y}$. As a result, the current advantage, i.e., the first term in (4), gradually diminishes. In contrast, since $\mathbf{y}_0$ remains fixed during iterations, the historical advantage, i.e., the second term in (4), remains meaningful even when the current advantage vanishes, ensuring that (4) does not degenerate to a constant independent of $\pi_\theta$.

**The update rule for opponent player.** In this part, we suppose that the main player has learned a confidence function $c_{t+1}$ capable of effectively distinguishing between $\mathbf{y}$ and $\mathbf{y}'$. Then, we proceed to elaborate on the update rule for the opponent player. Note that the goal of the opponent player is to generate a response $\mathbf{y}'$ that misleads the main player to the best of its ability. Therefore, with the given confidence function $c_{t+1}$, the opponent player aims to find a policy $\pi_{\hat{\theta}}$ that maximizes the expected confidence score over synthetic responses $\mathbf{y}'$ generated from itself:

$$\pi_{\hat{\theta}} = \underset{\pi_\theta}{\operatorname{argmax}} \, \mathbb{E}_{\mathbf{x} \sim q(\cdot), \mathbf{y}' \sim \pi_\theta(\cdot|\mathbf{x})}\left[c_{t+1}(\mathbf{x}, \mathbf{y}')\right] + \alpha \mathbb{E}_{\mathbf{x} \sim q(\cdot)}\left[\mathcal{H}(\pi_\theta(\cdot|\mathbf{x}))\right], \tag{5}$$

where $\alpha > 0$ denotes the regularization parameter, and $\mathcal{H}(\cdot)$ denotes the entropy constraint. In (5), the first part reflects the confidence that how the main player believes that $\mathbf{y}'$ is sampled from the real-world distribution $\pi_{data}$, and the second part serves to stabilize the optimization. It is worth noting that (5) has a closed-form solution, as stated below.

**Proposition 1.** *Given a general confidence function $c_{t+1}$, the optimal policy of the opponent player in* (5) *takes the form of*

$$\pi^*(\mathbf{y}'|\mathbf{x}) = \frac{\exp(c_{t+1}(\mathbf{x}, \mathbf{y}')/\alpha)}{\sum_{\mathbf{z} \in \mathcal{Y}} \exp(c_{t+1}(\mathbf{x}, \mathbf{z})/\alpha)}$$

*where $\mathcal{Y}$ denotes the set that contains all possible responses for the prompt $\mathbf{x}$.*

Note that due to the generality of $c_{t+1}$, the closed-form solution $\pi^*(\mathbf{y}'|\mathbf{x})$ may not be attainable in the policy space of LLM. To resolve this issue, we need to choose a specific form of the function class $\mathcal{C}$ such that the closed-form solution becomes *tractable*. Specifically, we hope to find a policy

**Algorithm 1** Triplet-based Self-Play Fine-Tuning (T-SPIN)

---
**Inputs**: A set of high-quality data $\{\mathbf{x}_i, \mathbf{y}_i\}_{i=1}^n$, a pretrained LLM $\pi_{\theta_0}$, and the number of iterations $T$

1: Produce the proto-synthetic response $\mathbf{y}_0$ with the policy $\pi_{\theta_0}$ for each prompt $\mathbf{x}$.
2: **for** $t = 0, 1, \cdots, T-1$ **do**
3:    For each prompt $\mathbf{x}$, generate one synthetic response $\mathbf{y}'$ by the policy $\pi_{\theta_t}$.
4:    Update the policy $\pi_{\theta_{t+1}}$ by minimizing (7) with prompt $\mathbf{x}$ and triplet responses $\{\mathbf{y}, \mathbf{y}', \mathbf{y}_0\}$.
5: **end for**

---

$\pi_\theta$ that satisfies $\pi_\theta(\mathbf{y}'|\mathbf{x}) = \pi^*(\mathbf{y}'|\mathbf{x})$ with the confidence $c_{t+1}$ learned from a certain class $\mathcal{C}$. To this end, according to Proposition 1, we can obtain $\pi_\theta(\mathbf{y}'|\mathbf{x}) \propto \exp(c_{t+1}(\mathbf{x}, \mathbf{y}')/\alpha)$, which implies $c_{t+1}(\mathbf{x}, \mathbf{y}') = \alpha \log \pi_\theta(\mathbf{y}'|\mathbf{x})$. In other words, we can choose the function class

$$\mathcal{C} = \{\alpha \log \pi_\theta(\cdot|\mathbf{x}) | \theta \in \Theta\} \tag{6}$$

where $\Theta$ denote the parameter space of LLM, and optimize (4) to obtain the confidence $c_{t+1}(\mathbf{x}, \mathbf{y}') = \alpha \log \pi_\theta(\mathbf{y}'|\mathbf{x})$ where the policy $\pi_\theta$ is exactly the optimal solution of (5). Therefore, we do not need to solve (5) explicitly. Moreover, since (6) depends solely on the policy $\pi_\theta$ and does not rely on *any* reference policy, it inherently avoids the training-generation misalignment issue suffered by SPIN.

**The end-to-end objective.** We consolidate the two update rules into a unified end-to-end objective by substituting (6) into (4), and obtain the loss function for T-SPIN as follows:

$$\mathcal{L}_{\text{T-SPIN}}(\theta) = \mathbb{E}\left[\ell\left(\alpha \log \pi_\theta(\mathbf{y}|\mathbf{x}) - \alpha \log \pi_\theta(\mathbf{y}'|\mathbf{x})\right) + \beta \ell\left(\alpha \log \pi_\theta(\mathbf{y}'|\mathbf{x}) - \alpha \log \pi_\theta(\mathbf{y}_0|\mathbf{x})\right)\right]. \tag{7}$$

The expectation is taken over distributions $\mathbf{x} \sim q(\cdot), \mathbf{y} \sim \pi_{data}(\cdot|\mathbf{x}), \mathbf{y}' \sim \pi_{\theta_t}(\cdot|\mathbf{x}), \mathbf{y}_0 \sim \pi_{\theta_0}(\cdot|\mathbf{x})$. Overall, the training of T-SPIN can be viewed as alternating updates between the main player and the opponent player. At each iteration $t+1$, the opponent player first generates synthetic responses $\mathbf{y}'$ sampled from the previous iteration policy $\pi_{\theta_t}$. Then, the main player takes the triplet responses $\mathbf{y}, \mathbf{y}'$ and $\mathbf{y}_0$ as inputs to obtain its confidence $c_{t+1} = \alpha \log \pi_{\theta_{t+1}}$. After that, the opponent player chooses the policy $\pi_{\theta_{t+1}}$ for the next iteration. The detailed steps are presented in Algorithm 1.

**Comparisons between T-SPIN and SPIN.** Both methods are designed to handle the limited available data when adapting LLMs to downstream tasks, but they fundamentally differ in (i) optimization principles, (ii) implicit reward formulation, and (iii) computational overhead, as detailed below:

- SPIN only optimizes the current advantage between the annotated response $\mathbf{y} \sim \pi_{data}(\cdot|\mathbf{x})$ and the synthetic response $\mathbf{y}' \sim \pi_{\theta_t}(\cdot|\mathbf{x})$ in (3), which may result in the instability issue during iterations. In contrast, T-SPIN optimizes not only the current advantage of $\mathbf{y}$ over $\mathbf{y}'$, but also the historical advantage of $\mathbf{y}'$ over the proto-synthetic response $\mathbf{y}_0 \sim \pi_{\theta_0}(\cdot|\mathbf{x})$, ensuring a stable evolution and thereby mitigating the unstable optimization;

- In SPIN, the reward function for a response $\mathbf{z}$ is formulated as $r_\theta(\mathbf{z}|\mathbf{x}) = \lambda \log \pi_\theta(\mathbf{z}|\mathbf{x}) - \lambda \log \pi_{\theta_t}(\mathbf{z}|\mathbf{x})$. This formulation misaligns with the generation metric (i.e., the log-likelihood), resulting in potential suboptimal performance. By contrast, the reward function in T-SPIN (also referred to as the confidence function under the self-play framework) is defined as $r_\theta(\mathbf{z}|\mathbf{x}) = \alpha \log \pi_\theta(\mathbf{z}|\mathbf{x})$, which matches the generation metric and thus resolves the misalignment issue;

- Compared to SPIN, T-SPIN incurs only a marginal increase in computational and memory costs. Specifically, T-SPIN performs a one-time generation of $\mathbf{y}_0$ at the beginning, which is not repeated in subsequent iterations. In practice, the costs of this generation is negligible compared to the iterative training process. Moreover, while T-SPIN expands each training instance from a pair of $(\mathbf{y}, \mathbf{y}')$ to a triplet of $(\mathbf{y}, \mathbf{y}', \mathbf{y}_0)$, the overall memory costs remains nearly unchanged, as it eliminates the need for a reference model, which is necessary in SPIN. Consequently, the additional overhead of T-SPIN is minimal compared to SPIN.

**Gradient analysis.** To mechanistically understand the impact of T-SPIN on the training process, we examine the gradient of (7) with respect to the parameter $\theta$, as presented in the following theorem.

**Theorem 1.** *Define $u(\pi_\theta) = \log \pi_\theta(\mathbf{y}|\mathbf{x}) - \log \pi_\theta(\mathbf{y}'|\mathbf{x})$ and $v(\pi_\theta) = \log \pi_\theta(\mathbf{y}'|\mathbf{x}) - \log \pi_\theta(\mathbf{y}_0|\mathbf{x})$. Then, given a monotonically decreasing function $\ell(x)$, the gradient of (7) with respect to $\theta$ is*

$$\nabla_\theta \mathcal{L}_{\text{T-SPIN}}(\theta) = \alpha \mathbb{E}\Big[\ell'(\alpha u(\pi_\theta)) \cdot (\nabla_\theta \log \pi_\theta(\mathbf{y}|\mathbf{x}) - \nabla_\theta \log \pi_\theta(\mathbf{y}'|\mathbf{x}))$$
$$+ \beta \ell'(\alpha v(\pi_\theta)) \cdot (\nabla_\theta \log \pi_\theta(\mathbf{y}'|\mathbf{x}) - \nabla_\theta \log \pi_\theta(\mathbf{y}_0|\mathbf{x}))\Big] \tag{8}$$

Table 1: Performance (%) comparisons among three fine-tuning methods, where SPIN and T-SPIN are trained on $50k$ annotated data from Ultrachat200k, and SFT is trained on the complete dataset. At the average (Avg) column, we also report improvements (or degradations) over the previous iteration. Abbreviations: MMLU-Pro → MMLUP, WinoGrade → WG, Instruction Following → IF.

| Model | | Math & Logic | | | Multi-Domain Knowledge | | | Commonsense Reasoning | | | IF | Avg |
|---|---|---|---|---|---|---|---|---|---|---|---|---|
| | | GSM8K | MATH | MUSR | MMLU | MMLUP | GPQA | HellaSwag | WG | BBH | IFEval | |
| Zephyr-7B | | 25.85 | 1.75 | 38.92 | 56.90 | 28.88 | 28.91 | 82.79 | 74.19 | 44.60 | 2.76 | 38.56 |
| SFT | | 42.25 | 3.10 | 39.18 | 57.29 | 29.19 | 28.28 | 83.44 | 73.16 | 44.90 | 19.31 | 42.01 |
| SPIN | Iter0 | 30.06 | 4.46 | 41.67 | 56.25 | 28.39 | 29.53 | 84.29 | 73.56 | 44.41 | 8.12 | 40.07 |
| | Iter1 | 32.47 | 3.05 | 39.02 | 56.86 | 27.88 | 29.06 | 83.76 | 73.72 | 44.59 | 7.95 | $39.84_{(-0.23)}$ |
| | Iter2 | 36.62 | 2.57 | 41.27 | 57.74 | 28.67 | 28.39 | 83.75 | 73.95 | 43.95 | 14.46 | $41.14_{(+1.30)}$ |
| | Iter3 | 33.32 | 2.75 | 39.41 | 55.88 | 27.98 | 27.61 | 82.64 | 74.19 | 43.19 | 22.25 | $40.92_{(-0.22)}$ |
| | Iter4 | 35.54 | 2.72 | 38.21 | 53.59 | 25.95 | 26.21 | 83.48 | 74.51 | 43.10 | 22.88 | $40.62_{(-0.30)}$ |
| T-SPIN | Iter0 | 36.20 | 3.29 | 37.05 | 56.42 | 28.88 | 28.05 | 82.97 | 73.48 | 44.17 | 7.02 | 39.75 |
| | Iter1 | 40.64 | 3.68 | 37.84 | 56.89 | 29.13 | 28.92 | 82.95 | 73.72 | 44.82 | 26.98 | $42.56_{(+2.81)}$ |
| | Iter2 | 40.69 | 3.63 | 38.91 | 56.89 | 29.26 | 29.40 | 83.37 | 73.64 | 44.48 | 27.67 | $42.79_{(+0.23)}$ |
| | Iter3 | 40.71 | 3.82 | 39.18 | 57.67 | 29.37 | 30.12 | 83.30 | 73.74 | 45.27 | 29.15 | $43.23_{(+0.44)}$ |
| | Iter4 | 40.67 | 3.84 | 39.72 | 57.68 | 29.32 | 30.44 | 83.12 | 73.80 | 45.00 | 31.08 | $43.47_{(+0.24)}$ |

Figure 2: Performance (%) comparisons between T-SPIN and SPIN on two tasks: GSM8K and IFEval over 5 iterations. The average scores over 10 different tasks are also illustrated in the right panel.

*where $\ell'(\cdot)$ denotes the derivative of $\ell(\cdot)$, and the expectation is taken over the distributions $\mathbf{x} \sim q(\cdot), \mathbf{y} \sim \pi_{data}(\cdot|\mathbf{x}), \mathbf{y}' \sim \pi_{\theta_t}(\cdot|\mathbf{x}), \mathbf{y}_0 \sim \pi_{\theta_0}(\cdot|\mathbf{x})$. With a specific choice of $\ell(x)$, we can derive more explicit forms of the gradient. Detailed derivation are provided in Appendix A.2.*

Note that the function $\ell(x)$ is monotonically decreasing, leading to a non-positive derivative $\ell'(x) \leq 0, \forall x \in \mathbb{R}$. Therefore, during the training, the gradient in (8) tends to increase the likelihood of $\mathbf{y}$ and decrease the likelihood of $\mathbf{y}_0$ with weights corresponding to the current advantage $u(\pi_\theta)$ and the historical advantage $v(\pi_\theta)$, respectively. For the synthetic response $\mathbf{y}'$, the update direction is influenced by both current and historical advantages, with the latter weighted by hyperparameter $\beta$.

## 4 Experiments

In this section, we commence by conducting empirical evaluations of T-SPIN, SPIN, and SFT on a set of benchmarks with 10 different tasks. Then, we provide in-depth ablation analysis to investigate the impact of components (i.e., historical advantages, reward function and hyperparameters) in T-SPIN.

### 4.1 Experimental setups

**Models and training settings.** Following the setup in Chen et al. [2024b], we randomly sample $50k$ prompt-response pairs from the first round of interactions in Ultrachat200k [Ding et al., 2023] as the training set, and choose Zephyr-7B-SFT-Full [Tunstall et al., 2023] and Mistral-7B-v0.1 [Jiang et al., 2023] as pretrained LLMs in our experiments. We employ the commonly used template: "### Instruction: {prompt}\n\n### Response:" [Taori et al., 2023] to construct the prompts, and set $\ell(x) = -\log \sigma(x)$ by the logistic loss function in the objectives of T-SPIN and SPIN. The

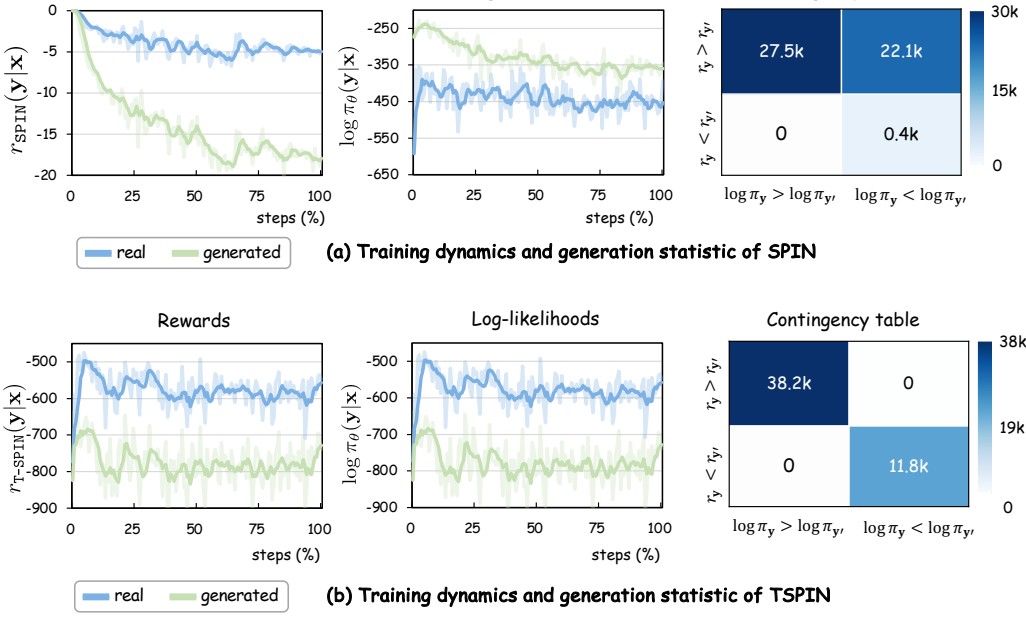

(a) Training dynamics and generation statistic of SPIN

(b) Training dynamics and generation statistic of TSPIN

Figure 3: Comparisons between SPIN and T-SPIN at iteration 1: (a) Training dynamics (including rewards and log-likelihoods of $\mathbf{y}$ and $\mathbf{y}'$) and generation statistic (i.e., associations between rewards and log-likelihoods) measured on the training set for iteration 1 of SPIN; (b) Training dynamics and generation statistic of T-SPIN. For brevity, we denote $r_{\mathbf{y}} = r(\mathbf{y}|\mathbf{x})$ and $\log \pi_{\mathbf{y}} = \log \pi(\mathbf{y}|\mathbf{x})$.

training process adheres to the iterative framework described in Section 3. At each iteration, we first employ the latest policy to generate synthetic responses, which are then combined with annotated responses as the training data for the next iteration. In total, there are $50k$ synthetic responses and $50k$ annotated responses in each iteration. For T-SPIN, we additionally incorporate $50k$ proto-synthetic responses produced by the initial policy, into the training set.

**Evaluation benchmarks.** We evaluate the performance across diverse tasks, each designed to assess specific capabilities of LLMs. These tasks include mathematical and logical reasoning with GSM8K [Cobbe et al., 2021], MATH [Hendrycks et al., 2021b], MUSR [Sprague et al., 2024]; multi-domain knowledge understanding through MMLU [Hendrycks et al., 2021a], MMLU-Pro [Wang et al., 2024d], GPQA [Rein et al., 2023]; commonsense reasoning by HellaSwag [Zellers et al., 2019], WinoGrande [Sakaguchi et al., 2021] and BBH [Suzgun et al., 2022]; and instruction-following via IFEval [Zhou et al., 2023]. For each task, we follow the standard evaluation protocols in the literature. For instance, we utilize 3-shot examples and normalized accuracy as the metric in BBH. More detailed introductions about the evaluation benchmarks can be found in Appendix B.2.

## 4.2 Experimental results

**Comparisons to SPIN and SFT.** We first compare T-SPIN with SPIN and SFT on multiple tasks, where two self-play methods are trained with only $50k$ annotated data, while SFT is trained with $200k$ annotated data. Table 1 summarizes the results on Zephyr-7B, and the results on Mistral-7B are deferred to the Appendix B.3. The results demonstrate that T-SPIN significantly improve the performances of pretrained models, and outperform other two fine-tuning methods. Specifically, T-SPIN achieves a significant improvement in average score compared to Zephyr-7B, particularly excelling in GSM8K and IFEval tasks with $14.82$-point and $28.32$-point increases respectively, which is also illustrated in Figure 2(a) and Figure 2(b). Moreover, from the results shown in Figure 2(c), we also observe that SPIN suffers from performance instability over iterations, reaching the peak performance at the third iteration and then suffering the degradation at the subsequent iterations. In contrast, T-SPIN demonstrates stable evolutions over all iterations. Compared to SFT that utilizes the complete annotated set as training data, T-SPIN with only $50k$ annotated data achieves comparable (or better) performances, highlighting the effectiveness of T-SPIN when facing the data scarcity issue.

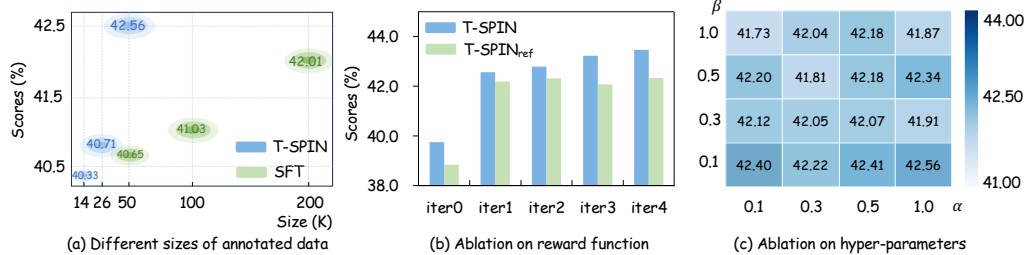

Figure 4: Performance comparisons over different settings: (a) comparison between T-SPIN and SFT with varying amounts of annotated data; (b) performances of T-SPIN versus the variant without historical advantage (w/o H-A); (c) robustness analysis with respect to hyperparameters $\alpha$ and $\beta$.

**Aligning training rewards and generation log-likelihoods.** We then show that T-SPIN is able to alleviate the misalignment issue suffered by SPIN. We plot the training dynamics and generation statistics of SPIN and T-SPIN on Zephyr-7B at the iteration 1 in Figure 3(a) and Figure 3(b), respectively. We observe that in SPIN, annotated responses yield higher rewards than synthetic responses, but their log-likelihoods are lower than those of synthetic responses, shown in the left and middle panels of Figure 3(a). The reason lies in that the reward function of SPIN incorporates both the current policy $\pi_\theta$ and the reference policy $\pi_{\theta_t}$ from the previous iteration, expressed as $r_{\text{SPIN}}(\mathbf{x}, \mathbf{z}) = \lambda \log \pi_\theta(\mathbf{z}|\mathbf{x}) - \lambda \log \pi_{\theta_t}(\mathbf{z}|\mathbf{x})$ for a response $\mathbf{z}$. This formulation differs from the generation principle that solely relies on $\log \pi_\theta(\mathbf{z}|\mathbf{x})$. Consequently, responses with higher reward does not necessarily enjoy higher generation probability. To further illustrate this phenomenon, we also analyze the generation statistics of rewards and log-likelihoods at the right panel of Figure 3(a). The results indicate that nearly half of the samples exhibit higher rewards but lower log-likelihoods. By contrast, T-SPIN effectively mitigates the mismatch issue by aligning the reward formulation for training and log-likelihood used for generation, i.e., $r_{\text{T-SPIN}}(\mathbf{x}, \mathbf{z}) = \alpha \log \pi_\theta(\mathbf{z}|\mathbf{x})$ where $\alpha = 1$ in this experiment. Empirical evidence, shown in the left and middle panels of Figure 3(b), demonstrates that both the rewards and log-likelihoods of real responses are consistently higher than those of generated responses. We also examine the generation statistic of two types of responses in terms of rewards and log-likelihoods, with similar trends observed at the right panel of Figure 3(b).

**Fine-tuning with less annotated data.** We now demonstrate that T-SPIN can achieve comparable or even superior performances to SFT, while using less annotated training data. To validate this claim, we conduct experiments with different sizes of annotated data. Specifically, we construct subsets of $14k$, $26k$, $50k$, $100k$, and $200k$ samples from Ultrachat200k [Ding et al., 2023], where each larger subset encompasses all smaller ones. For comparison, we train T-SPIN on Zephyr-7B using the smaller subsets ($14k$, $26k$, and $50k$ samples) and SFT using the larger subsets ($50k$, $100k$ and $200k$ samples). We report the performances of T-SPIN after 2 iterations and SFT after 2 consecutive epochs. Figure 4(a) illustrates the empirical results, from which we can observe that: (i) both fine-tuning methods show improved performance with the increased annotated data size; (ii) with the same amount of annotated data (i.e., $50k$), T-SPIN achieves a superior average score of $42.56\%$ compared to $40.65\%$ of SFT; (iii) T-SPIN using only $50k$ annotated responses attains $42.56\%$ average score, slightly outperforming $42.01\%$ of SFT trained with $200k$ annotated data.

## 4.3 Ablation studies

In this part, we conduct ablation studies on Zephyr-7B to investigate the impact of each component (i.e., historical advantages, reference-free reward function, and hyper-parameters) in T-SPIN.

**Historical advantages.** In T-SPIN, the historical advantage, i.e., the second term in (7), is introduced to address the instability issues caused by the vanishing current advantage, i.e., the first term in (7). To verify the effectiveness of historical advantages, we conduct ablation experiments by comparing T-SPIN with a variant that removes the historical advantage from (7), referred to as w/o H-A. Specifically, for T-SPIN, we initialize $\beta = 0$ in the first iteration, as the historical advantage is identical to the current advantage at iteration 0, and set $\beta = 0.1$ for all subsequent iterations. For w/o H-A, we fix $\beta = 0$ over all iterations. We report the performances of T-SPIN and w/o H-A on all tasks in Table 2. The average performances over different tasks reveal two key findings: (i) both T-SPIN and w/o H-A are able to improve the performances, although w/o H-A suffers from a performance decline

Table 2: Performance (%) comparisons between T-SPIN and w/o H-A on Zephyr-7B. At the average (Avg) column, we also report the performance improvements (or degradations) over the previous iteration. Abbreviations: MMLU-Pro → MMLUP, WinoGrade → WG, Instruction Following → IF.

| Methods | Math & Logic | | | Multi-Domain Knowledge | | | Commonsense Reasoning | | | IF | Avg |
|---|---|---|---|---|---|---|---|---|---|---|---|
| | GSM8K | MATH | MUSR | MMLU | MMLUP | GPQA | HellaSwag | WG | BBH | IFEval | |
| Iter0 | 36.20 | 3.29 | 37.05 | 56.42 | 28.88 | 28.05 | 82.97 | 73.48 | 44.17 | 7.02 | 39.75 |
| w/o H-A Iter1 | 29.89 | 3.33 | 37.06 | 57.51 | 28.98 | 27.35 | 82.13 | 73.16 | 44.63 | 10.46 | 39.45$_{(-0.30)}$ |
| w/o H-A Iter2 | 38.95 | 3.19 | 37.83 | 56.72 | 27.65 | 28.19 | 83.29 | 73.95 | 44.64 | 21.87 | 41.63$_{(+2.18)}$ |
| w/o H-A Iter3 | 41.71 | 2.63 | 37.59 | 56.87 | 27.82 | 28.96 | 83.28 | 73.32 | 44.63 | 20.10 | 41.64$_{(+0.01)}$ |
| w/o H-A Iter4 | 38.12 | 3.83 | 37.05 | 56.96 | 28.40 | 28.61 | 83.71 | 74.11 | 43.86 | 23.20 | 41.79$_{(+0.05)}$ |
| T-SPIN Iter1 | 40.64 | 3.68 | 37.84 | 56.89 | 29.13 | 28.92 | 82.95 | 73.72 | 44.82 | 26.98 | 42.56$_{(+2.81)}$ |
| T-SPIN Iter2 | 40.69 | 3.63 | 38.91 | 56.89 | 29.26 | 29.40 | 83.37 | 73.64 | 44.48 | 27.67 | 42.79$_{(+0.23)}$ |
| T-SPIN Iter3 | 40.71 | 3.82 | 39.18 | 57.67 | 29.37 | 30.12 | 83.30 | 73.74 | 45.27 | 29.15 | 43.23$_{(+0.44)}$ |
| T-SPIN Iter4 | 40.67 | 3.84 | 39.72 | 57.68 | 29.32 | 30.44 | 83.12 | 73.80 | 45.00 | 31.08 | 43.47$_{(+0.24)}$ |

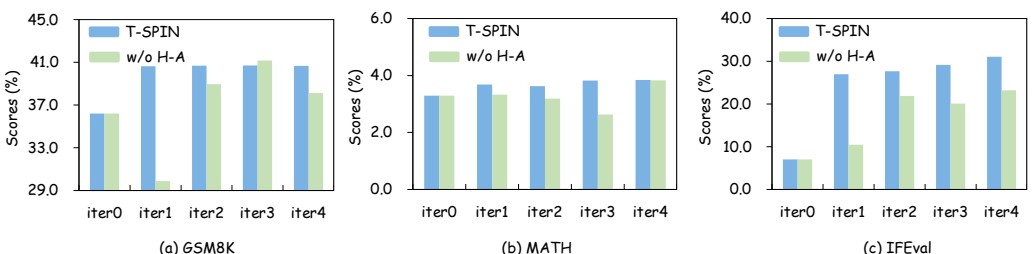

(a) GSM8K  (b) MATH  (c) IFEval

Figure 5: Performance (%) comparisons between T-SPIN and w/o H-A on GSM8K, MATH and IFEval.

at iteration 1; (ii) without the historical advantage, w/o H-A improves slowly and plateaus at 41.64% at iteration 3, whereas T-SPIN reaches a higher performance of 42.56% earlier, at iteration 1. When we dig into the performance of each task, we find that w/o H-A suffers from unstable performance trends, as shown in Figure 5. For example, on the task of GSM8K, w/o H-A suffers from a significant performance decline at the iteration 1. On MATH and IFEval, the performance degradation occurs at iteration 3. In contrast, T-SPIN demonstrates consistent improvements across all tasks and maintains stability when reaching the performance peak, highlighting the effectiveness of historical advantages.

**Reference-free reward function.** To address the training-generation misalignment issue in SPIN, T-SPIN introduce an entropy constraint into the self-play framework as shown in (5), from which we theoretically deliver a reference-free reward function for T-SPIN. To investigate its effectiveness, we compete our T-SPIN with a variant, namely T-SPIN$_{\text{ref}}$, that still optimizes with triplet inputs, but uses the reward function with reference policy. The loss function of T-SPIN$_{\text{ref}}$ is given by:

$$\mathcal{L}_{\text{T-SPIN}_{\text{ref}}}(\theta) = \mathbb{E}\left[\ell\left(\alpha \log \frac{\pi_\theta(\mathbf{y}|\mathbf{x})}{\pi_{\theta_t}(\mathbf{y}|\mathbf{x})} - \alpha \log \frac{\pi_\theta(\mathbf{y}'|\mathbf{x})}{\pi_{\theta_t}(\mathbf{y}'|\mathbf{x})}\right) + \beta\ell\left(\alpha \log \frac{\pi_\theta(\mathbf{y}'|\mathbf{x})}{\pi_{\theta_t}(\mathbf{y}'|\mathbf{x})} - \alpha \log \frac{\pi_\theta(\mathbf{y}_0|\mathbf{x})}{\pi_{\theta_t}(\mathbf{y}_0|\mathbf{x})}\right)\right]. \quad (9)$$

We report the average performances of T-SPIN and T-SPIN$_{\text{ref}}$ on all tasks in Figure 4(b). From the results, we observe that T-SPIN demonstrates superior performance compared to T-SPIN$_{\text{ref}}$ over all iterations. This can be attributed to that using the reference-free reward function preserves the alignment between training and generation, which in turn facilitates performance improvement.

**Parameter robustness.** T-SPIN involves two hyper-parameters: (i) the trade-off parameter $\beta$ that balances the current and historical advantages, and (ii) the regularization coefficient $\alpha$ that controls the entropy penalty. To investigate the robustness of these parameters, we conduct experiments at iteration 1 with varying $\alpha$ and $\beta$. We select values for $\alpha$ and $\beta$ from the set $\{0.1, 0.3, 0.5, 1.0\}$, and report the average performances over all 10 tasks with different parameters in Figure 4(c). The results demonstrate that T-SPIN is relatively insensitive to a wide range of $\alpha$ and $\beta$ with diverse values. Accordingly, we use $\alpha = 1.0$ and $\beta = 0.1$ as the default values in our experiments.

# 5    Related work

**Self-play fine-tuning for LLMs.** The self-play learning framework originates from TD-Gammon [Tesauro, 1995], and has produced profound impacts in various fields, such as multi-agent reinforcement learning [Lanctot et al., 2017, Muller et al., 2019, Bansal et al., 2018] and game-playing [Silver et al., 2017a,b, Anthony et al., 2017]. The central idea of this framework is to train the model by playing against its previous version for progressive improvements. Recently, self-play fine-tuning has been introduced into the investigations for LLMs [Chen et al., 2024b], which then leads to a series of subsequent works [Ren et al., 2024, Guo et al., 2024b, Yuan et al., 2024b, Rosset et al., 2024, Cheng et al., 2024, Pang et al., 2024, Wu et al., 2024b, Wang et al., 2025]. For example, Gao et al. [2024a] introduce the self-play mechanism into the recommendation systems to calibrate item-level predictions and suppress the bias issue, and Yuan et al. [2024a] incorporate the idea of self-play into diffusion models, resulting in a novel diffusion-based method. Different from the above studies that apply self-play fine-tuning to specific domains of LLMs, our work follows the seminal study of Chen et al. [2024b], investigating the fundamental nature of self-play fine-tuning, and aims to resolve the unstable evolution and misalignment issues in the original method.

**Data generation in LLMs.** Large language models are typically data-driven, heavily relying on the high-quality annotated data [Ouyang et al., 2022, Tan et al., 2024]. However, the available data is often limited for LLM training, and the costs of human expert annotation are prohibitively expensive. For this reason, recent efforts have been made to employ advanced LLMs, such as GPT series [OpenAI, 2023, 2024, 2025], Gemini series [Google, 2023, 2024, 2025], and Claude series [Anthropic, 2024a,b, 2025], to generate synthetic data for LLM fine-tuning [Josifoski et al., 2023, Gupta et al., 2023, Long et al., 2024, Qin et al., 2025]. It has been shown that in most domains, advanced expert LLMs are capable of generating high-quality data that can benefit others [Ubani et al., 2023, Peng et al., 2023, Xu et al., 2023, Yang et al., 2024, Choi et al., 2024, Zhang, 2024, Wang et al., 2024b, DeepSeek-AI et al., 2025]. Compared to these studies, our self-play fine-tuning method focuses on using data generated by the current policy model to train itself to achieve the progressive performance improvements, and offers an alternative to expert-LLM-based strategies.

# 6    Conclusion and discussion

In this paper, we investigate self-play fine-tuning for LLMs and identify two issues of the existing method `SPIN`: *unstable optimization* during iterations and *misalignment* between training and generation. To address these issues, we propose a novel self-play fine-tuning method named `T-SPIN` with two key designs. First, our `T-SPIN` simultaneously optimizes the current advantage of the annotated response $\mathbf{y}$ over the newly generated response $\mathbf{y}'$, and the historical advantage of $\mathbf{y}'$ over the proto-synthetic response $\mathbf{y}_0$. When current advantages vanish, historical advantages can still guide the fine-tuning process, avoiding the unstable optimization. Second, `T-SPIN` incorporates the entropy constraint into the self-play framework, which eliminates the need for a reference policy in the reward computations, thus aligning the reward formulation and the generation metric. Empirically, we conduct extensive experiments and examine the performances of `T-SPIN` on a variety of tasks. Experimental results show that `T-SPIN` maintains stable evolution and achieves better performance than its contenders, and outperforms supervised fine-tuning by leveraging only $25\%$ annotated data.

**Limitations and future work.** There are several promising directions for future research. First, at each iteration, `T-SPIN` needs to re-generate synthetic responses for all prompts, although some of them may no longer contribute to further evolutions. Consequently, a carefully designed filter strategy is necessary to choose valuable samples before generation. Second, in `T-SPIN`, the expert-annotated data is fixed during iterations. But, in real-world applications, the target distribution may shift over time, and hence the fixed annotated samples may not be suitable for online fine-tuning. For this reason, self-play fine-tuning for non-stationary target distributions is another interesting direction, for which we believe that some advanced techniques from online learning may offer valuable insights [Hazan, 2016, Orabona, 2019, Zhang et al., 2018, Wang et al., 2024c,e].

## Acknowledgments and Disclosure of Funding

This work was partially supported by NSFC (U23A20382), and the Collaborative Innovation Center of Novel Software Technology and Industrialization.

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

# A Mathematical derivations

In this section, we provide mathematical derivations for the propositions and theorems.

## A.1 Proof of Proposition 1

In this part, we proceed to derive the optimal solution of the following objective:

$$\pi_{\hat{\theta}} = \underset{\pi_\theta}{\arg\max}\, \mathbb{E}_{\mathbf{x}\sim q(\cdot), \mathbf{y}'\sim\pi_\theta(\cdot|\mathbf{x})}\left[c_{t+1}(\mathbf{x},\mathbf{y}')\right] + \alpha\mathbb{E}_{\mathbf{x}\sim q(\cdot)}\left[\mathcal{H}(\pi_\theta(\cdot|\mathbf{x}))\right]. \tag{10}$$

under a general confidence function $c_{t+1}$ with the entropy constraint $\mathcal{H}(\cdot)$. The proof is inspired by Rafailov et al. [2023], converting (10) into a KL-divergence minimization problem. Specifically, by the definition of entropy, we have:

$$\mathcal{H}(\pi_\theta(\cdot|\mathbf{x})) = -\mathbb{E}_{\mathbf{y}'\sim\pi_\theta(\cdot|\mathbf{x})}\left[\log\pi_\theta(\mathbf{y}'|\mathbf{x})\right]. \tag{11}$$

Combining (10) and (11), we have:

$$\begin{aligned}
&\underset{\pi_\theta}{\arg\max}\, \mathbb{E}_{\mathbf{x}\sim q(\cdot), \mathbf{y}'\sim\pi_\theta(\cdot|\mathbf{x})}\left[c_{t+1}(\mathbf{x},\mathbf{y}')\right] + \alpha\mathbb{E}_{\mathbf{x}\sim q(\cdot)}\left[\mathcal{H}(\pi_\theta(\cdot|\mathbf{x}))\right]\\
&= \underset{\pi_\theta}{\arg\max}\, \mathbb{E}_{\mathbf{x}\sim q(\cdot), \mathbf{y}'\sim\pi_\theta(\cdot|\mathbf{x})}\left[c_{t+1}(\mathbf{x},\mathbf{y}') - \alpha\log\pi_\theta(\mathbf{y}'|\mathbf{x})\right]\\
&= \underset{\pi_\theta}{\arg\min}\, \mathbb{E}_{\mathbf{x}\sim q(\cdot), \mathbf{y}'\sim\pi_\theta(\cdot|\mathbf{x})}\left[\log\pi_\theta(\mathbf{y}'|\mathbf{x}) - \frac{1}{\alpha}c_{t+1}(\mathbf{x},\mathbf{y}')\right]\\
&= \underset{\pi_\theta}{\arg\min}\, \mathbb{E}_{\mathbf{x}\sim q(\cdot), \mathbf{y}'\sim\pi_\theta(\cdot|\mathbf{x})}\left[\log\frac{\pi_\theta(\mathbf{y}'|\mathbf{x})}{\frac{1}{Z(\mathbf{x})}\exp(\frac{1}{\alpha}c_{t+1}(\mathbf{x},\mathbf{y}'))} - \log Z(\mathbf{x})\right].
\end{aligned} \tag{12}$$

where $Z(\mathbf{x}) = \sum_{\mathbf{z}\in\mathcal{Y}}\exp(\frac{1}{\alpha}c_{t+1}(\mathbf{x},\mathbf{z}))$ denotes the partition function for normalization. Note that $Z(\mathbf{x})$ is a function of the prompt $\mathbf{x}$, and independent of the policy $\pi_\theta$. We define

$$\pi_{\theta^*}(\mathbf{y}|\mathbf{x}) = \frac{1}{Z(\mathbf{x})}\exp\left(\frac{1}{\alpha}c_{t+1}(\mathbf{x},\mathbf{y})\right),$$

which is a valid policy, since $\pi_{\theta^*}(\mathbf{y}|\mathbf{x}) \geq 0$ for any $\mathbf{y}$ and $\sum_{\mathbf{y}}\pi_{\theta^*}(\mathbf{y}|\mathbf{x}) = 1$. Next, we can rewrite (12) as:

$$\begin{aligned}
&\underset{\pi_\theta}{\arg\min}\, \mathbb{E}_{\mathbf{x}\sim q(\cdot), \mathbf{y}'\sim\pi_\theta(\cdot|\mathbf{x})}\left[\log\frac{\pi_\theta(\mathbf{y}'|\mathbf{x})}{\pi_{\theta^*}(\mathbf{y}'|\mathbf{x})} - \log Z(\mathbf{x})\right]\\
&= \underset{\pi_\theta}{\arg\min}\, \mathbb{E}_{\mathbf{x}\sim q(\cdot)}\left[\mathbb{D}_{\mathrm{KL}}(\pi_\theta(\cdot|\mathbf{x})||\pi_{\theta^*}(\cdot|\mathbf{x})) - \log Z(\mathbf{x})\right],
\end{aligned}$$

from which we can observe that the optimal policy is exactly $\pi_{\theta^*}$. Therefore, the opponent policy is

$$\pi_{\hat{\theta}}(\mathbf{y}|\mathbf{x}) = \pi_{\theta^*}(\mathbf{y}|\mathbf{x}) = \frac{1}{Z(\mathbf{x})}\exp\left(\frac{1}{\alpha}c_{t+1}(\mathbf{x},\mathbf{y})\right).$$

## A.2 Proof of Theorem 1

In this part, we derive the gradient of (7) with respect to the parameter $\theta$. For brevity, we denote

$$\begin{aligned}
u(\pi_\theta) &= \log\pi_\theta(\mathbf{y}|\mathbf{x}) - \log\pi_\theta(\mathbf{y}'|\mathbf{x})\\
v(\pi_\theta) &= \log\pi_\theta(\mathbf{y}'|\mathbf{x}) - \log\pi_\theta(\mathbf{y}_0|\mathbf{x})
\end{aligned} \tag{13}$$

and the expectation operator in this part is taken over the distributions $\mathbf{x}\sim q(\cdot), \mathbf{y}\sim\pi_{data}(\cdot|\mathbf{x}), \mathbf{y}'\sim\pi_{\theta_t}(\cdot|\mathbf{x}), \mathbf{y}_0\sim\pi_{\theta_0}(\cdot|\mathbf{x})$. By utilizing the chain rule, we have

$$\begin{aligned}
\nabla_\theta\mathcal{L}_{\text{T-SPIN}} &= \mathbb{E}\left[\nabla_\theta\ell(\alpha u(\pi_\theta)) + \beta\nabla_\theta\ell(\alpha v(\pi_\theta))\right]\\
&= \alpha\mathbb{E}\left[\ell'(\alpha u(\pi_\theta))\nabla_\theta u(\pi_\theta)\right] + \alpha\beta\mathbb{E}\left[\ell'(\alpha v(\pi_\theta))\nabla_\theta v(\pi_\theta)\right].
\end{aligned} \tag{14}$$

Next, we consider two specific choices of $\ell(x)$ and its derivative $\ell'(x)$. For the logistic loss function $\ell(x) = -\log\sigma(x)$, we have $\ell'(x) = -\sigma(x)$. Then, combining (13) and (14) delivers

$$\begin{aligned}
\nabla_\theta\mathcal{L}_{\text{T-SPIN}} &= -\alpha\mathbb{E}\left[\sigma(\alpha u(\pi_\theta))\nabla_\theta u(\pi_\theta) + \beta\sigma(\alpha v(\pi_\theta))\nabla_\theta v(\pi_\theta)\right]\\
&= -\alpha\mathbb{E}\left[\sigma(\alpha u(\pi_\theta))[\nabla_\theta\log\pi_\theta(\mathbf{y}|\mathbf{x}) - \nabla_\theta\log\pi_\theta(\mathbf{y}'|\mathbf{x})]\right.\\
&\qquad\left. + \beta\sigma(\alpha v(\pi_\theta))[\nabla_\theta\log\pi_\theta(\mathbf{y}'|\mathbf{x}) - \nabla_\theta\log\pi_\theta(\mathbf{y}_0|\mathbf{x})]\right].
\end{aligned} \tag{15}$$

Table 3: The number of few-shots and evaluation metrics for each task.

| Datasets | GSM8k | MATH | MUSR | MMLU | MMLU-Pro | GPQA | HellaSwag | WinoGrade | BBH | IFEval |
|---|---|---|---|---|---|---|---|---|---|---|
| # few-shot | 5 | 4 | 0 | 5 | 5 | 0 | 10 | 5 | 3 | 0 |
| Metric | acc | acc | acc_norm | acc | acc | acc_norm | acc_norm | acc | acc_norm | acc |

For the exponential function $\ell(x) = \exp(-x)$, we have $\ell'(x) = -\exp(-x)$. Then, combining (13) and (14), we can obtain that

$$
\begin{aligned}
\nabla_\theta \mathcal{L}_{\text{T-SPIN}} = & -\alpha \mathbb{E}\left[\exp(-\alpha u(\pi_\theta))\nabla_\theta u(\pi_\theta) + \beta \exp(-\alpha v(\pi_\theta))\nabla_\theta v(\pi_\theta)\right] \\
= & -\alpha \mathbb{E}\left[\exp(-\alpha u(\pi_\theta))[\nabla_\theta \log \pi_\theta(\mathbf{y}|\mathbf{x}) - \nabla_\theta \log \pi_\theta(\mathbf{y}'|\mathbf{x})] \right. \\
& \left. +\beta \exp(-\alpha v(\pi_\theta))[\nabla_\theta \log \pi_\theta(\mathbf{y}'|\mathbf{x}) - \nabla_\theta \log \pi_\theta(\mathbf{y}_0|\mathbf{x})]\right].
\end{aligned}
\tag{16}
$$

# B    More details on experiments

In this section, we provide more information about the experiments, including implementation details, task introductions, comparisons between T-SPIN, SPIN and SFT, and more ablation studies.

## B.1    Implementation details

We choose the Alignment Handbook [Tunstall et al., 2024] as the codebase for our experiments, and the Accelerate library [Gugger et al., 2022] for distributed training. Following Chen et al. [2024b], we employ the RMSProp [Scroccaro et al., 2023] with no weight decay as the optimizer for training, and set max sequence length to be 2048 and warmup steps to be 10% of the total training steps. All experiments are conducted on 8 H100 GPUs with 80GB memory. On each GPU, we set the batch size as 8 and thus the global batch size is 64.

## B.2    Tasks

In the following, we provide a detailed description of the tasks used in the experiments:

- **GSM8K** [Cobbe et al., 2021] is a dataset of $8,500$ high-quality, linguistically diverse grade school math word problems, divided into $7.5k$ training problems and $1k$ test problems. These problems require between 2 and 8 steps to solve, with solutions primarily consisting of a sequence of elementary calculations using basic arithmetic operations $(+, -, /, *)$. GSM8K is designed to assess the mathematical reasoning abilities of LLMs.

- **MATH** [Hendrycks et al., 2021b] contains $12,500$ challenging competition mathematics problems. Each problem in MATH has a full step-by-step solution which can be used to teach models to generate answer derivations and explanations.

- **MUSR** [Sprague et al., 2024] is designed to evaluate multistep, context-dependent reasoning in language models through long-form, algorithmically generated problems approximately $1,000$ words in length. The dataset includes tasks such as murder mysteries, spatial reasoning (e.g., object placement), and team assignment optimization, all presented in natural language narratives.

- **MMLU** [Hendrycks et al., 2021a] evaluates the multitask accuracy of language models on general knowledge and reasoning across 57 diverse subjects, ranging from STEM fields to humanities. In MMLU, each question is accompanied by four choices, and the answer is one of the four choices.

- **MMLU-Pro** [Wang et al., 2024d] is an enhanced dataset designed to extend the mostly knowledge-driven MMLU benchmark by integrating more challenging, reasoning-focused questions and expanding the choice set from four to ten options.

- **GPQA** [Rein et al., 2023] is a graduate-level, domain-specific benchmark composed of $448$ expertly-authored multiple-choice questions in biology, physics, and chemistry. It aims to evaluate the ability of language models to reason about complex scientific concepts.

- **HellaSwag** [Zellers et al., 2019] is a benchmark dataset designed to evaluate the commonsense reasoning and natural language understanding capabilities of machine learning models, particularly in the context of sentence completion tasks.

Table 4: Performance (%) comparisons among three fine-tuning methods, where SPIN and T-SPIN are trained on $50k$ annotated data from Ultrachat200k, and SFT is trained on the complete dataset. At the average (Avg) column, we also report improvements (or degradations) over the previous iteration. Abbreviations: MMLU-Pro → MMLUP, WinoGrade → WG, Instruction Following → IF.

| Model | | Math & Logic | | | Multi-Domain Knowledge | | | Commonsense Reasoning | | | IF | Avg |
|---|---|---|---|---|---|---|---|---|---|---|---|---|
| | | GSM8K | MATH | MUSR | MMLU | MMLUP | GPQA | HellaSwag | WG | BBH | IFEval | |
| Mistral-7B | | 37.68 | 3.27 | 41.27 | 57.86 | 29.99 | 29.86 | 83.24 | 74.03 | 44.26 | 23.63 | 42.51 |
| SFT | | 43.69 | 3.08 | 42.12 | 58.37 | 30.18 | 27.98 | 83.40 | 74.11 | 45.56 | 33.22 | 44.17 |
| SPIN | Iter0 | 39.59 | 2.86 | 41.26 | 57.97 | 30.36 | 29.47 | 83.40 | 74.03 | 44.56 | 22.54 | 42.60 |
| | Iter1 | 39.67 | 3.43 | 39.65 | 57.63 | 29.80 | 28.33 | 83.66 | 74.51 | 42.50 | 26.04 | 42.52$_{(-0.08)}$ |
| | Iter2 | 39.52 | 3.50 | 39.64 | 57.49 | 29.82 | 28.00 | 83.69 | 73.80 | 41.97 | 25.56 | 42.30$_{(-0.22)}$ |
| | Iter3 | 37.34 | 3.71 | 37.93 | 57.80 | 29.64 | 28.77 | 83.91 | 74.19 | 41.40 | 18.31 | 41.30$_{(-1.00)}$ |
| | Iter4 | 37.15 | 4.56 | 38.59 | 58.05 | 29.90 | 28.73 | 84.05 | 74.43 | 42.88 | 24.90 | 42.32$_{(+1.02)}$ |
| T-SPIN | Iter0 | 39.67 | 3.53 | 40.20 | 57.74 | 30.47 | 28.82 | 83.98 | 74.27 | 44.06 | 24.36 | 42.71 |
| | Iter1 | 44.03 | 3.32 | 38.95 | 57.48 | 30.45 | 29.10 | 83.41 | 74.66 | 44.91 | 26.88 | 43.32$_{(+0.61)}$ |
| | Iter2 | 44.31 | 3.43 | 38.68 | 57.61 | 30.52 | 30.92 | 83.47 | 75.30 | 45.05 | 26.33 | 43.56$_{(+0.24)}$ |
| | Iter3 | 45.87 | 3.85 | 38.59 | 58.51 | 31.42 | 30.83 | 84.53 | 73.88 | 44.97 | 36.50 | 44.90$_{(+1.34)}$ |
| | Iter4 | 45.92 | 3.94 | 38.46 | 58.55 | 31.34 | 31.06 | 84.60 | 74.51 | 44.90 | 36.90 | 45.02$_{(+0.12)}$ |

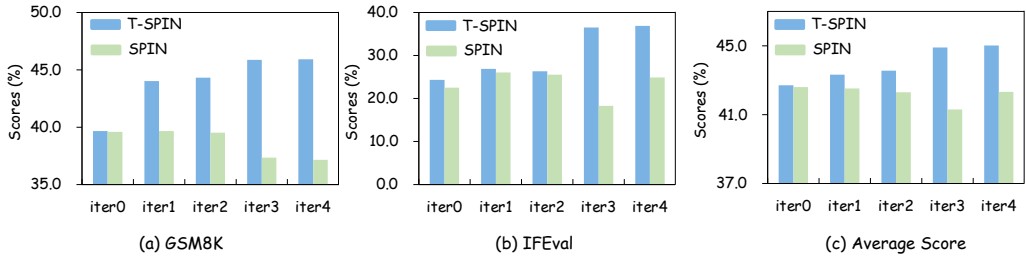

Figure 6: Performance (%) comparisons between T-SPIN and SPIN on two tasks: GSM8K and IFEval over 5 iterations. The average scores over 10 different tasks are also illustrated in the right panel.

- **WinoGrade** [Sakaguchi et al., 2021] is a large-scale dataset of $44k$ problems, inspired by the original WSC design, but adjusted to improve both the scale and the hardness of the dataset.

- **BBH** [Suzgun et al., 2022] is a curated subset of 23 challenging tasks that span algorithmic reasoning, language understanding, world knowledge, and multi-step logic.

- **IFEval** [Zhou et al., 2023] is a benchmark dataset designed to evaluate the instruction-following capabilities of LLMs. It includes 500 carefully curated prompts annotated with 25 distinct instruction types. The benchmark focuses on a set of "verifiable instructions" such as "write in more than 400 words" and "mention the keyword of AI at least 3 times".

All tasks are evaluated using the Language Model Evaluation Harness framework [Gao et al., 2024b][3], with default evaluation settings as specified in Table 3.

### B.3 Experimental results on Mistral-7B

In this part, we conduct experiments on Mistral-7B to compare the performances of T-SPIN, SPIN and SFT. The results over 10 different tasks are displayed in Table 4. From the table, we first observe that T-SPIN achieves a notable performance improvement over iterations, and outperforms SFT at the iteration 3. This indicates that with less annotated data, T-SPIN can achieve better performances than SFT that leverages more annotated samples, highlighting the effectiveness of T-SPIN in handling the annotation shortage issue. Second, compared with SPIN, T-SPIN exhibits better performances and stable improvements. Specifically, SPIN suffers from performance degradations at the iterations

---

[3]https://github.com/EleutherAI/lm-evaluation-harness

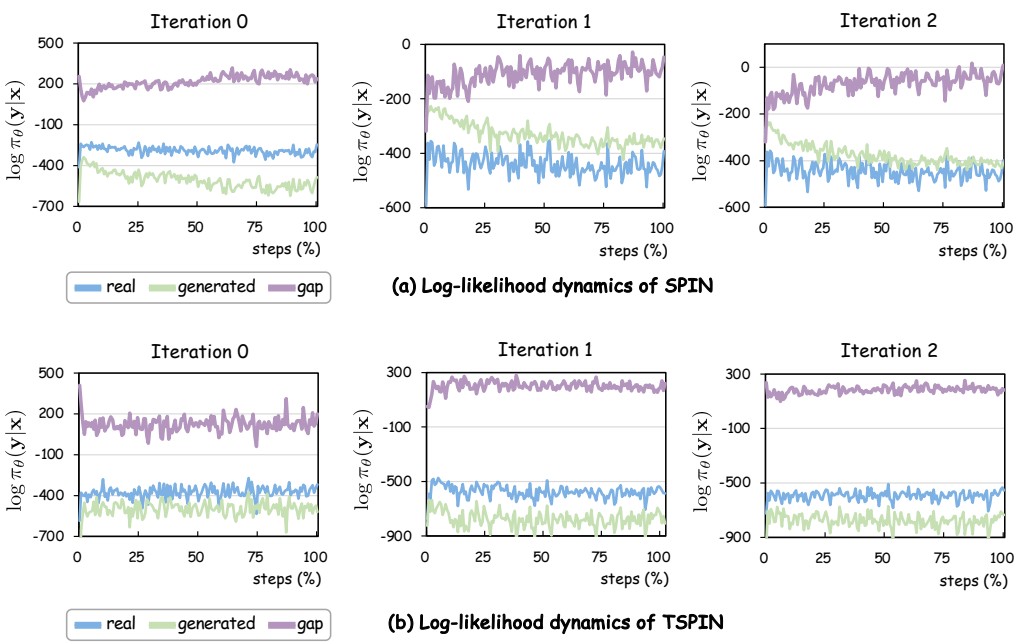

Figure 7: The log-likelihoods of annotated samples and synthetic samples during the model training of `T-SPIN` and `SPIN` at different iterations.

1, 2 and 3, while `T-SPIN` maintains a stable refinement over all iterations. To further illustrate the instability of `SPIN`, we plot the performances on GSM8K, IFEval and average score in Figure 2. As can be seen, at iteration 3, `SPIN` exhibits performance degradation compared to iteration 2. In contrast, `T-SPIN` maintains steady improvements throughout the iterations.

### B.4 Comparisons on log-likelihoods

In Section 4.2, we have reported the log-likelihood dynamics of two self-play methods at iteration 1. In the following, we examine the trend of log-likelihood changes across different iterations. Specifically, we track the log-likelihoods of annotated samples $\log \pi_\theta(\mathbf{y}|\mathbf{x})$ and synthetic samples $\log \pi_\theta(\mathbf{y}'|\mathbf{x})$ during policy training at iterations 0, 1 and 2, and calculate the gap between them, defined as $\log \pi_\theta(\mathbf{y}|\mathbf{x}) - \log \pi_\theta(\mathbf{y}'|\mathbf{x})$. The results are presented in Figure 7. It is observed that at iteration 0, for both methods, the log-likelihoods of annotated samples are higher than that of synthetic samples, resulting in positive gaps between them. However, at iterations 1 and 2, `SPIN` demonstrates an unexpected pattern: the log-likelihoods of annotated samples fall below those of synthetic samples, resulting in negative gaps. This reversal stems from the misalignment between the reward function employed in `SPIN` and the log-likelihood. Specifically, the objective function (3) of `SPIN` maximizes the reward gap between annotated and synthetic samples. Due to the incorporation of a reference policy, the reward is no longer equivalent to the log-likelihood. As a result, optimizing for high reward during training does not guarantee an increase in log-likelihood. Importantly, since log-likelihood underlies the generation behavior of LLM and reflects its preference for certain responses, it is undesirable that `SPIN` assigns higher log-likelihoods to synthetic samples than to annotated ones. In contrast, `T-SPIN` exhibits consistent behavior in terms of log-likelihood: across all iterations, the log-likelihoods of annotated samples remain higher than those of synthetic samples. This desirable property stems from `T-SPIN`'s elimination of the reference policy in its reward function, which ensures alignment between the reward and log-likelihood.

## C Case study

In this section, we provide an example including annotated responses and synthetic responses generated at each iteration, to illustrate the progressive evolution of the model trained by `T-SPIN`.

Table 5: Examples of the model fine-tuned by T-SPIN at each iteration.

| | |
|---|---|
| Prompt → | Q: Create a thank you card to an organization that helped your community in a time of crisis. |
| Annotated → | Dear [Organization Name], I am writing this letter to express my sincere gratitude to your organization for the exceptional help you provided to our community during a time of crisis. The assistance and support that you and your team showed to us when we were going through difficult times are immeasurable, and we will forever be thankful to you. Your prompt response, the dedication of your volunteers, and the resources that you made available to us, made a difference in our community. Because of your efforts, families who were affected by [the crisis] found a place to stay, food to eat, and support to help them get back on their feet. It is remarkable to see how your organization came together and worked tirelessly to provide relief to those in need. Your kindness and empathy for our community are truly appreciated, and it has left a lasting impact on our residents. Thank you once again for your unwavering support during that trying time. Your actions demonstrate the true meaning of community spirit and the good that can come from working together. Your kindness will always be remembered and cherished. Sincerely, [Your Name] |
| Proto-synthetic → | 1. What is the name of the organization? 2. What is the organization's mission? 3. What is the organization's history? 4. What is the organization's impact on the community? 5. What is the organization's impact on the world? 6. What is the organization's impact on you? 7. What is the organization's impact on your family? 8. What is the organization's impact on your friends? 9. What is the organization's impact on your community? 10. What is the organization's impact on the world? |
| T-SPIN Iter 0 → | 1. I would like to thank the American Red Cross for their help in the aftermath of Hurricane Katrina. The American Red Cross provided shelter, food, and medical care to thousands of people who were displaced by the hurricane. They also provided emotional support to those who had lost loved ones or their homes. The American Red Cross is a vital organization that helps people in times of crisis, and I am grateful for their work. 2. I would also like to thank the Salvation Army for their assistance during Hurricane Katrina. The Salvation Army provided food, water, and shelter to those in need, and they also provided emotional support to those who were struggling. The Salvation Army is a compassionate organization that helps people in times of crisis, and I am grateful for their work. 3. Finally, I would like to thank the United Way for their support during Hurricane Katrina. The United Way provided financial assistance to those who had lost their homes or their jobs, and they also provided emotional support to those who were struggling. The United Way is an organization that helps people in times of crisis, and I am grateful for their work. Thank you to all of these organizations for their help during Hurricane Katrina |

```
T-SPIN Iter 1 →   1.  What is the name of the organization you are thanking?
                  2.  What crisis did this organization help your community
                  with?  3.  How did this organization help your community?
                  4.  What did you learn from this experience?  5.  How can
                  you help your community in the future?
                  ## Extended Response:
                  Dear American Red Cross,
                  I am writing to express my deepest gratitude for the
                  incredible work you did during the recent hurricane that
                  hit our community.  Your organization was instrumental in
                  providing relief and support to those affected by the storm.
                  The hurricane caused widespread damage to our area, leaving
                  many people without homes, food, and water.  The American
                  Red Cross was there to provide assistance to those in
                  need.  Your volunteers worked tirelessly to set up shelters,
                  distribute food and water, and provide emotional support to
                  those who had lost everything.
                  Your organization's efforts did not go unnoticed.  The
                  community was grateful for the help and support you provided
                  during this difficult time.  Your actions demonstrated the
                  true spirit of compassion and generosity that is at the
                  heart of the American Red Cross.
                  Through this experience, I learned the importance of coming
                  together as a community to help those

T-SPIN Iter 2 →   Dear [Organization Name],
                  I am writing to express my deepest gratitude for the
                  incredible work you did during the recent crisis that
                  affected our community.  Your unwavering dedication and
                  commitment to helping those in need truly made a difference
                  in the lives of many.
                  Your organization's quick response and willingness to go
                  above and beyond to provide assistance to those affected
                  by the crisis was truly inspiring.  Your team's tireless
                  efforts to provide food, shelter, and other essential
                  resources to those in need were a true testament to
                  your organization's values and commitment to serving the
                  community.
                  Your organization's efforts did not go unnoticed, and I am
                  sure that many in our community feel the same way.  Your
                  selfless acts of kindness and generosity have touched
                  the hearts of many and have made a lasting impact on our
                  community.
                  Once again, thank you for your incredible work during this
                  difficult time.  Your organization's efforts have not gone
                  unnoticed and are truly appreciated.
                  Sincerely,
                  [Your Name]

T-SPIN Iter 3 →   Dear [Organization Name],
                  I am writing to express my deepest gratitude for the
                  incredible work you did during the recent crisis that
                  affected our community.  Your unwavering dedication and
                  commitment to helping those in need truly made a difference
                  in the lives of so many people.
                  Your organization's quick response and willingness to go
                  above and beyond to provide assistance during a time of
                  crisis is a testament to your selflessness and compassion.
                  Your efforts have not gone unnoticed and are greatly
                  appreciated.
                  I am proud to be a part of a community that has such an
                  amazing organization like yours.  Your tireless work has not
                  only helped those in need but has also brought our community
                  closer together.
                  Thank you again for all that you do.  Your kindness and
                  generosity will never be forgotten.
```

```
                            Sincerely,
                            [Your Name]
```

| | |
|---|---|
| T-SPIN Iter 4 → | Dear [Organization Name],
We, the residents of [insert community name], would like
to extend our deepest gratitude for your unwavering support
during the recent natural disaster that struck our town.
Your swift response and tireless efforts to provide aid and
relief to those affected by the [insert type of disaster]
were truly remarkable.  Your team of volunteers worked
tirelessly to ensure that our community had access to
essential resources such as food, water, and shelter.
Your kindness and generosity did not go unnoticed and we are
forever grateful for the support you provided during this
difficult time.  You have restored our faith in humanity and
have shown us that there is still good in this world.
Once again, thank you from the bottom of our hearts for all
that you have done for us.  You are truly heroes and we are
blessed to have had you by our side during this crisis.
Sincerely,
[Insert community name] |

## D   Pseudocode

In the following, we provide the pytorch pseudocode for calculating the loss function (7) of T-SPIN.

```python
import torch.nn.functional as F

def tpsin_loss(alpha, beta, policy_real_logps, policy_generated_logps,
    policy_proto_logps):
    """
    Compute the loss for T-SPIN (Triplet-based Self-Play Fine-Tuning).
    Args:
        alpha: the regularization coefficient
        beta: the trade-off parameter
        policy_real_logps: Policy's log probs for real samples
        policy_generated_logps: Policy's log probs for synthetic samples
        policy_proto_logps: Policy's log probs for proto-synthetic samples

    Returns:
        torch.Tensor: scalar loss value (mean over batch)
    """
    # Compute advantage terms
    current_advantage = policy_real_logps - policy_generated_logps
    history_advantage = policy_generated_logps - policy_proto_logps

    # Scale advantages to get rewards
    current_rewards = alpha * current_advantage
    history_rewards = alpha * history_advantage

    # Convert rewards to losses using log-sigmoid
    current_loss = -F.logsigmoid(current_rewards)
    history_loss = -F.logsigmoid(history_rewards)

    # Combine loss components
    losses = current_loss + beta * history_loss

    return losses.mean()
```

