# OpenReview forum: "Triplets Better Than Pairs: Towards Stable and Effective Self-Play Fine-Tuning for LLMs"
_NeurIPS.cc/2025/Conference — NeurIPS 2025 poster_

### Official Review · Reviewer_wsFg · 2025-06-15

**Clarity:** 3
**Significance:** 3
**Originality:** 4
**Rating:** 4
**Confidence:** 5

**Summary:**

This paper proposes an incremental improvement over a well known RLHF method called SPIN. SPIN and T-SPIN proposed in this paper both leverage the same format of data as SFT, and the comparison is constrained to this setting.

SPIN loss looks very similar to DPO loss except that, because we don't have the losing answer for query, the model's latest version's generation is used for that purpose.

**Questions:**

One of the selling point seems to be the stability of the method. If so, an experiment comparing the variance of the training process would be great.

**Ethical Concerns:**

["NO or VERY MINOR ethics concerns only"]

**Final Justification:**

I read the authors' response and find it convincing, especially the explanations on the the reference.

**Limitations:**

It's still in the realm of SFT-data utilization, which is understandable for a fair experimental comparison.

**Quality:**

3

**Strengths And Weaknesses:**

Strength:
- T-SPIN proposes to use some sort of triplet comparison to not only encourage its previous generated are thought of as inferior than annotation, but additionally prefer that over the initial reference answer. The experimental results look convincing.


Weakness:
- It was on older models and doesn't compare with preference-pair-based methods (which is understandable, but a table would be informative).
- Isn't the reference still playing a role in the loss function? Instead of living in the denominator, it now resides in the second term of comparison.
- One of the points the author's critique of SPIN is that the reward is not the same as metric. However, isn't it still the case in T-SPIN?

---

> ### Author Rebuttal · Authors · 2025-07-30
>
> We sincerely appreciate the valuable feedback! We provide our responses as follows, and hope the reviewer could **reassess** our work. We are looking forward to addressing any further question in the reviewer-author discussion period!
>
> ---
>
> **Q1**: It was on older model, and do not compare to preference-pair-based methods.
> **A1**:  Thanks for the helpful suggestion. We present our responses as follows.
>
> * **LLM choice**. We conduct experiments on Zephyr-7B and Mistral-7B to remain consistent with the original settings in SPIN. Furthermore, we have conducted additional experiments on Qwen2.5-7B-base, and evaluated performance on ten benchmarks (in Lines 218-223). We report the average score in the following table:
>
>   ||Qwen 2.5-7B|iter0|iter1|iter2|iter3|iter4|
>   | --------| -------------| -------| -------| -------| -------| -------|
>   |SPIN|53.49|53.84|53.92|53.71|53.89|53.98|
>   |T-SPIN|53.49|53.68|54.17|55.04|54.97|55.03|
>
>   From the results, we observe that: First, both methods yield less performance enhancements on Qwen2.5-7B compared to Zephyr‑7B and Mistral‑7B. This may be attributed to: (i) Qwen2.5 is already a strong LLM, making it harder for further improvements via fine-tuning; and (ii) our experiments are conducted with limited annotated samples, where achieving performance enhancements on strong LLMs is inherently challenging. Second, SPIN still suffers from unstable optimization during iterations, which prevents further performance improvements. In contrast, T-SPIN introduces historical advantages to guide the optimization, resulting in stable performance enhancements.
>
> * **Comparison to preference-pair methods.**  We would like to emphasize that our study focuses on the problem of *self-play fine-tuning*, which is fundamentally different from *pairwise preference optimization*. Specifically, preference optimization typically relies on data in the format of (prompt, chosen response, rejected response), while our T-SPIN is designed for scenarios where only SFT-style data, i.e., (prompt, high-quality annotated response) is available. Our setting is more common in practice, as SFT-style data is more widely available than preference pairs. Therefore, a direct comparison between self-play fine-tuning and preference optimization may not be appropriate. Nevertheless, we sincerely appreciate the reviewer's helpful suggestion, and we will include a comparison with pairwise preference optimization methods in the revised version.
>
> ---
>
> **Q2**: Isn't the reference still playing a role in the loss function?
>
> **A2**: Thanks for the valuable question. We feel that there are some misunderstandings regarding the term '*reference'* . In our work, *reference* specifically refers to the *reference policy*. SPIN is considered reference-dependent because its objective (3) explicitly relies on a reference policy $\pi_{\theta_t}$. In contrast, T-SPIN is reference-free, as its loss function (7) depends solely on the current policy $\pi_\theta$ and **does not involve any other policies** during optimization. We will include a detailed clarification in the revised version.
>
> ---
>
> **Q3**: The reward function in T-SPIN.
>
> **A3**: In SPIN, the reward function is defined as the log-ratio $c(x, y)=\lambda \log \frac{\pi\_{\theta}(y|x)}{\pi_{\theta_t}(y|x)}$ between the current model $\pi_\theta$ and a reference model $\pi_{\theta_t}$. According to (3), SPIN aims to maximize the reward for annotated samples while minimizing that for synthetic ones. However, in SPIN, the reward function used for training is not aligned with the metric $\log p_{\theta}(y|x)$ used by LLMs for generation. In other words, a response with a high reward does not necessarily correspond to a high generation probability, which has been verified in **the contingency table of Figure 3(a)** . In contrast, T-SPIN adopts the reward function $c(x, y)=\alpha \log  \pi_{\theta}(y|x)$ for training, matching that used for generation. As a result, samples with higher rewards are also more likely to be generated, as shown in **the contingency table of Figure 3(b)** . We sincerely thank the reviewer for raising the question and will include a clear explanation in the revised version.
>
> ---
>
> **Q4**: An experiment comparing the variance of the training process.
>
> **A4**: Thank you for the constructive suggestion. In our paper, *stability* refers to the consistent improvement of model performance over iterations, without suffering from performance collapse. In contrast, the reviewer seems to refer to the stability of *the training process itself*, which differs from our intended focus. We will offer a detailed clarification in the revised version. Moreover, additional experiments about training process variance will also be included.

---

### Official Review · Reviewer_J2Mt · 2025-07-02

**Clarity:** 4
**Significance:** 3
**Originality:** 3
**Rating:** 5
**Confidence:** 4

**Summary:**

This paper proposes Triplet-based Self-Play Fine-tuning (T-SPIN). It is an improved method for fine-tuning large language models with limited annotated data. The authors identify two key limitations in the existing SPIN method: (1) unstable optimization when the advantage between annotated and synthetic responses diminishes over iterations, and (2) misalignment between training rewards and generation metrics due to the use of reference policies. The proposed method, T-SPIN, addresses these 2 issues through two main innovations: (1) incorporating "historical advantages" by comparing current synthetic responses to proto-synthetic responses from the initial policy, and (2) eliminating reference policies by introducing an entropy constraint.

**Questions:**

How does T-SPIN perform on datasets from different domains (e.g., code generation, scientific text, creative writing)?

**Ethical Concerns:**

["NO or VERY MINOR ethics concerns only"]

**Limitations:**

Yes

**Paper Formatting Concerns:**

No.

**Quality:**

3

**Strengths And Weaknesses:**

Strengths:
1. The paper effectively identifies the limitations of SPIN with both theoretical analysis and empirical evidence. The misalignment between training rewards and generation log-likelihoods is well-demonstrated.
2. The experiments cover diverse tasks (e.g., math, logic, knowledge, reasoning, instruction-following, etc.) and include analysis, like ablation studies, etc. The results consistently show T-SPIN's superiority over SPIN in both performance and stability. It shows that to achieve comparable performance to supervised fine-tuning only needs 25% of the annotated data. It is useful especially when we have limited expert annotations.

Weaknesses:
1. While the paper provides gradient analysis (e.g., in Theorem 1), it lacks deeper theoretical guarantees about convergence. The stability claims are primarily based on empirical studies rather than theoretically proven.
2. The proposed method, T-SPIN, requires generating and storing proto-synthetic responses from the initial policy. It will potentially increase memory and computational requirements. To my best knowledge, I didn't see the paper discussed somewhere about addressing the practical implications of this overhead.
3. Experiments are done with only on 7B parameter models (e.g., Zephyr-7B, Mistral-7B). It's unclear how the method scales to larger models where computational costs become more significant. Meanwhile, all experiments use Ultrachat200k. The generalizability of the performance on other domains remains unknown.

---

> ### Author Rebuttal · Authors · 2025-07-30
>
> We sincerely appreciate the valuable feedback! We provide our responses below, and are looking forward to addressing any further question in the reviewer-author discussion period!
>
> ---
>
> **Q1**: This paper lacks deeper theoretical guarantees about convergence. The stability claims are primarily based on empirical studies rather than theoretically proven.
>
> **A1**: Thank you for the valuable comment. We agree that establishing theorems about convergence and stability is important for T-SPIN. However, due to **the non-convex property of LLMs** and **the arbitrariness of the monotonically decreasing function** ***$\ell(\cdot)$*** in (7), it is highly challenging to provide rigorous convergence and stability guarantees for T-SPIN. As an alternative, we provide theoretical analysis for gradients in Theorem 1 with corresponding discussions in **Lines 196–200**. We recognize the importance of convergence guarantee and plan to investigate it in future work.
>
> ---
>
> **Q2**: Lack of discussion on memory and computational requirements.
>
> **A2**: Thank you for this practical concern. The additional cost of T-SPIN compared to SPIN comes from the proto-synthetic response $y_0$ generated by the initial policy $\pi_0$, including
>
> * **One-time computational cost:**  At the beginning of the training, we conduct a single forward pass to generate $y_0$ with the initial policy $\pi_0$ (see **Algorithm 1**). The computational cost is equivalent to generating synthetic response $y'$ at one iteration, but is not repeated during the training process. In practice, the generation of proto-synthetic responses only takes 0.8 hours, which is significantly lower than the 3.5 hours for training.
> * **Increased memory cost:**  In T-SPIN, the training data for each iteration expands from pairs ($y, y'$) to triplets ($y, y', y_0$). We argue that this is a modest and acceptable trade-off for the *significant* improvements in final performance and optimization stability, as shown in Table 1 and Figure 2. Moreover, compared to SPIN, although T-SPIN utilizes additional samples for training, it **eliminates the need of reference model**. As a result, the additional memory cost in experiments is minimal.
>
> We will include the discussion about computational and memory costs in the final version.
>
> ---
>
> **Q3**: Performance of larger models.
>
> **A3**: Thanks for the helpful suggestion. While our T-SPIN is model-agnostic,  we are currently unable to provide results on larger LLMs due to limited computational resources and the tight rebuttal timeline. Nevertheless, we plan to include expanded results in the final version.
>
> ---
>
> **Q4**: All experiments use Ultrachat200k. How does T-SPIN perform on datasets from different domains.
>
> **A4**: Thank you for the insightful question. Following the experimental setting in SPIN, we select a subset of Ultrachat200k as the training set. In our paper, we have evaluated the performances over a wide range of domains, including Math and Logic (GSM8K, MATH, MUSR), Multi-Domain Knowledge (MMLU, MMLU-Pro, GPQA), Commonsense Reasoning (HellaSwag, Winogrande, BBH), and Instruction Following (IFEval), and present the results in Table 1. Furthermore, we also evaluate the performance of T-SPIN on  **Code Generation** (HumanEval [1]) and **Scientific Question Answering** (OpenBookQA [2]). The results presented in the following table demonstrate that T-SPIN consistently yields stable performance improvements across these diverse benchmarks.
>
> ||Zephyr-7B|iter0|iter1|iter2|iter3|iter4|
> | --------------------| -----------| -------| -------| -------| -------| -------|
> |HumanEval (pass@1)|3.05|3.60|4.27|4.88|6.1|6.71|
> |OpenBookQA|42.8|43.0|43.4|43.8|44.2|44.4|
>
> ---
>
> **Reference**
>
> [1] Chen et al. Evaluating Large Language Models Trained on Code. Arxiv, 2021.
>
> [2] Mihaylov et al. Can a Suit of Armor Conduct Electricity? A New Dataset for Open Book Question Answering. EMNLP, 2018.

---

### Official Review · Reviewer_q2aJ · 2025-07-03

**Clarity:** 3
**Significance:** 3
**Originality:** 3
**Rating:** 5
**Confidence:** 3

**Summary:**

This paper introduces T-SPIN, which constructs a triplet of annotated, synthetic, and proto-synthetic responses to improve SPIN and stabilizes training by leveraging historical advantages and aligning the reward with generative likelihood through an entropy constraint. Extensive experiments on multiple aspects are done to compare T-SPIN and SPIN, showing that T-SPIN achieves comparable or better performance than SPIN.

**Questions:**

Have the authors considered using responses from an intermediate past policy, such as $\pi_{\theta_{\lfloor t/2 \rfloor}}$, instead of always relying on the initial policy $\pi_{\theta_0}$? This could provide a stronger and more relevant benchmark for $\pi_{\theta_t}$, potentially encouraging greater improvements in later iterations.

**Ethical Concerns:**

["NO or VERY MINOR ethics concerns only"]

**Final Justification:**

The authors addressed my concerns during our discussion and raised my score from 4 to 5.

**Limitations:**

Yes.

**Paper Formatting Concerns:**

No.

**Quality:**

3

**Strengths And Weaknesses:**

**Strengths**:

1.	Stabilizes the optimization by introducing historical advantages from proto-synthetic responses, preventing optimization collapse.
2.	Aligns the reward function with the generative likelihood.
3.	Experiment shows that there is a consistent improvement in the fine-tuned performance as iteration number increases.


**Weaknesses**:

1. The comparison with the proto-synthetic responses $y_0$ may become stale over successive iterations, as the model’s performance could surpass the initial policy to such an extent that this anchor no longer provides a meaningful training signal.

2. The ablation studies could be expanded. In particular, it would be valuable to analyze the isolated effect of changing the reward formulation from $\lambda \log \frac{\pi_\theta(y|x)}{\pi_{\theta_t}(y|x)}$ to $\alpha \log \pi_\theta(y|x)$, in order to disentangle the contribution from the historical advantage.

3. The paper does not clearly articulate why the historical advantage is helpful, despite this being a central and intriguing idea. A more thorough discussion of the underlying intuition and mechanisms behind the benefits of historical advantage would strengthen the work.

---

> ### Author Rebuttal · Authors · 2025-07-30
>
> We sincerely appreciate the valuable feedback! We provide our responses below, and are looking forward to addressing any further question in the reviewer-author discussion period!
>
> ---
>
> **Q1**: The proto-synthetic responses $y_0$ may become stale over successive iterations.
>
> **A1**: Thank you for the insightful comment. We understand your concern that proto-synthetic response $y_0$ may become stale over iterations. As shown in (7), $y_0$ is incorporated to compute the historical advantages, which  can be viewed as a regularization to prevent performance collapse when current advantages diminish during iterations. In fact, its static property is a key factor in providing stable optimization. Detailed discussions can also be found in **Lines 136-144**.
>
> ---
>
> **Q2**: The isolated effect of changing the reward formulation.
>
> **A2**: Thank your for the constructive suggestion. We currently employ the reward function $c(x, y)=\lambda \log \frac{\pi_{\theta}(y|x)}{\pi_{\theta_t}(y|x)}$ , and substitute it into (4) to obtain a new loss function, as shown below:
>
> $$
> L(\theta) =   \mathbb{E}  \left[ \ell\left( \lambda  \log  \frac{\pi_\theta (y |x )}{\pi_{\theta_t} (y |x )}    - \lambda  \log \frac{  \pi_{\theta}(y'|x)}{\pi_{\theta_{t}}(y'|x)}   \right)  + \beta \ell\left( \lambda  \log \frac{  \pi_{\theta}(y'|x)}{\pi_{\theta_t}(y'|x)}    - \lambda  \log \frac{  \pi_{\theta}(y_0|x)}{\pi_{\theta_t}(y_0|x)}    \right)  \right].
> $$
>
> Then, we conduct an ablation study on Zephyr-7B, comparing the performance of models optimized with (7) (i.e., T-SPIN) to those using the above loss function (namely, **T-SPIN_ref**). The average score over ten tasks is presented in the following table. From the results, we observe that *T-SPIN* demonstrates superior performance compared to *T-SPIN_ref.*  This can be attributed to the fact that using $c(x, y)=\alpha \log  \pi_{\theta}(y|x)$ as the reward function preserves the alignment between training and generation, which in turn facilitates performance improvement.
>
> ||Zephyr-7B|iter0|iter1|iter2|iter3|iter4|
> | ------------| -----------| -------| -------| -------| -------| -------|
> |T-SPIN_ref|38.56|38.83|42.19|42.32|42.08|42.33|
> |T-SPIN|38.56|39.75|42.56|42.79|43.23|43.47|
>
>
> ---
>
> **Q3**: A more thorough discussion of the historical advantage.
>
> **A3**: Thank you for the helpful suggestion. To clarify the underlying intuition, we start by examining the fundamental issue that causes the performance collapse observed in SPIN during iterative optimization.
>
> * **Vanishing current advantages.**  The core idea of SPIN is to progressively improve performance by minimizing the relative advantage of the current policy $\pi_{data}$ over the target policy $\pi_{\theta_t}$ during iterations. Specifically, as shown in (3), SPIN aims to reduce the confidence gap between high-quality annotated samples and self-generated responses, i.e., minimizing $\ell(c(x,y)-c(x,y'))$ where $y \sim \pi_{data}$ , $y'\sim \pi_{\theta_t} $ and $c(x,y)=\lambda \log \frac{\pi_\theta(y|x)}{\pi_{\theta_t}(y|x)}$.  As the model continues to evolve, the gap between $\pi_{data}$ and $\pi_{\theta_t}$ gradually narrows. Once this gap vanishes, the objective in (3) degenerates into a constant independent on $\pi_\theta$, **making any policy a trivial solution**, leading to unstable optimization and performance collapse ultimately.
>
> To stabilize the iterative optimization, it is thus necessary to introduce a regularization term that prevents this degeneration. A natural choice is to incorporate the **historical advantage**​, i.e., the relative advantage of the current policy $\pi_{\theta_t}$ over the initial policy $\pi_{\theta_0}$, which remains meaningful even when the current advantage vanishes, thereby mitigating objective degeneration and maintaining training stability. We will include a more in-depth discussion of the underlying intuition and mechanisms of T-SPIN in the revised version.
>
> ---
>
> **Q4**: Have the authors considered using responses from an intermediate past policy.
>
> **A4**: We sincerely appreciate this insightful suggestion! To evaluate its effectiveness, we conduct additional experiments on Zephyr-7B comparing our T-SPIN with the variant (namely **T-SPIN_mid**) that employs proto-synthetic responses $y_{\lfloor t/2 \rfloor}$ from the intermediate past policy $\pi_{\lfloor t/2 \rfloor}$ to optimize (7). Notably, at iterations 0 and 1, both T-SPIN and T-SPIN\_mid use proto-synthetic responses generated from the same initial policy $\pi_0$, so the comparison starts from iteration 2. We present the experimental results as follows:
>
> ||iter2|iter3|iter4|
> | --------------| -------| -------| -------|
> |T-SPIN_mid|42.27|42.20|41.53|
> |T-SPIN|42.79|43.23|43.47|
>
>
> From the table, we observe that T-SPIN\_mid suffers from the performance decline across iterations. We hypothesize that this degradation may stem from the vanishing of the historical advantage at iteration 2 (since both $y'$ and $y_0$ are sampled from $\pi_{\theta_1}$), which may potentially destabilize the optimization in subsequent iterations. Due to limited computational resources, we are unable to further validate this idea over other models during the rebuttal period. This is indeed a very **interesting** direction, and we plan to explore it in future work. Once again, we sincerely thank the reviewer for this valuable idea.

---

> > ### Comment · Reviewer_q2aJ · 2025-08-05
> >
> > Thanks for the detailed explanation and running additional experiments, which have addressed most of my concerns.
> >
> > I still have some confusion regarding the claim: *"the objective in (3) degenerates into a constant independent on $\pi_\theta$"*. I can imagine that the gradient might become flat as the iteration t increases, but why would the optimal solution become a constant independent of $\pi_\theta$?
> >
> > This has been asserted in the manuscript for multiple times. Is there a reference that discusses this phenomenon, or is it something first observed by the authors? Is there any evidence supporting this claim? Thank you!

---

> > > ### Author Response · Authors · 2025-08-05
> > >
> > > Thanks for the follow-up question. Our responses are provided below:
> > >
> > > * First, we would like to clarify that our statement is “**the objective** becomes a constant independent of $\pi_\theta$” not "**the optimal solution** becomes a constant independent of $\pi_\theta$".
> > >
> > >
> > > * Then, let us consider the loss function used in SPIN. Specifically, SPIN aims to minimize
> > > $$
> > > L_{\text{SPIN}}(\theta)=\mathbb{E}\left[\ell\left( \lambda \log \frac{\pi_\theta(y|x)}{\pi_{\theta_t}(y|x)}  - \lambda \log \frac{\pi_\theta(y'|x)}{\pi_{\theta_t}(y'|x)} \right) \right].
> > > $$
> > > When $y=y'$, the loss function degenerates to $L_{\text{SPIN}}(\theta) = \mathbb{E}\left[ \ell(0) \right] = C$, which is vacuous, as it is independent of the policy $\pi_\theta$, and any policy $\pi_{\hat{\theta}}$ can be the optimal solution. Consequently, further optimization on this loss function is meaningless, and leads to the potential performance degradation. Now, we explain why T-SPIN can handle this situation. The loss function of T-SPIN is
> > > $$
> > > L_{\text{T-SPIN}}(\theta)=\mathbb{E}\left[ \ell\left(\alpha \log \pi_\theta(y|x) - \alpha \log \pi_\theta(y'|x) \right) + \beta \ell\left(\alpha \log \pi_\theta(y'|x) - \alpha \log \pi_\theta(y_0|x) \right) \right].
> > > $$
> > > When $y=y'$, the first term vanishes and the loss function becomes $L_{\text{T-SPIN}}(\theta)=\mathbb{E}\left[ \beta \ell\left(\alpha \log \pi_\theta(y'|x) - \alpha \log \pi_\theta(y_0|x) \right) \right].$ This loss function still depends on the policy $\pi_\theta$, ensuring that it does not degenerate to a constant. The relevant discussion can be found in **Lines 140-144**.
> > >
> > >
> > > * Recently, we find that [1] also points out the instability issue of SPIN. However, [1] only provides empirical evidence, whereas we offer both theoretical analyses and experimental validations, and propose a novel self-play fine-tuning method T-SPIN.
> > >
> > >
> > > We hope our explanation can address your concerns, and we will provide a clearer clarification in the revised version. Once again, we sincerely appreciate your time and valuable feedback!
> > >
> > >
> > > ---
> > >
> > >
> > > **Reference.**
> > >
> > > [1] Alami et al. Investigating Regularization of Self-Play Language Models. Arxiv, 2024.

---

> > > > ### Comment · Reviewer_q2aJ · 2025-08-07
> > > >
> > > > Thanks for the detailed response, which has addressed my concerns.

---

> > > > > ### Author Response · Authors · 2025-08-07
> > > > >
> > > > > We appreciate your feedback and acknowledgment of our work, and will incorporate your valuable suggestions in the final version.

---

### Official Review · Reviewer_Xd9e · 2025-07-23

**Clarity:** 3
**Significance:** 2
**Originality:** 3
**Rating:** 4
**Confidence:** 2

**Summary:**

This paper proposes a triplet-based self-play method for fine-tuning LLMs called T-SPIN. Compared to SPIN, it uses triplets of annotated, synthetic, and proto-synthetic responses to address the deminishing advantage issue and replaces the reference‑policy reward with an entropy‑regularised objective. The authors test their method on Zephyr‑7B and Mistral‑7B over ten benchmarks and show consistent improvements over SPIN with better data efficiency.

**Questions:**

How many iterations do you observe diminishing returns?

**Ethical Concerns:**

["NO or VERY MINOR ethics concerns only"]

**Limitations:**

Please see "Weakness"

**Quality:**

3

**Strengths And Weaknesses:**

### Strengths
1. The paper is generally well-written and easy to read.
2. The triplet idea, which uses historical advantages to avoid collapses, can be novel.
3. The improvement on data efficiency (using 25% of labeled data) can be useful in practice. Training also seems more stable than SPIN.
4. The authors did some ablation studies to verify the effectiveness of historical advantages.

### Weakness
1. The models used for finetuning, Zephyr‑7B and Mistral‑7B, are under-performing modern models like the Qwen series. It is unclear if this method also holds for stronger baselines or if the improvement would be marginal. Results could be more convincing if the authors also test on stronger baselines.
2. The extra computational overhead of generating proto-synthetic responses is not measured, including wall-clock time, flops, and memory cost.
2. The improvement over SPIN is not significant on some datasets.
2. Not compared with other improved methods based off SPIN.

---

> ### Author Rebuttal · Authors · 2025-07-30
>
> We sincerely appreciate the valuable feedback, and provide our responses as follows. We hope our responses help strengthen your ***confidence*** in this work, and we are looking forward to addressing any further questions during the reviewer-author discussion.
>
> ---
>
> **Q1**: It is unclear if this method also holds for stronger baselines.
>
> **A1**: Thank your for the helpful suggestion. We have conducted additional experiments on Qwen2.5-7B-base, and evaluated performance on ten benchmarks (in Lines 218-223). We report the average score in the following table:
>
> ||Qwen 2.5-7B|iter0|iter1|iter2|iter3|iter4|
> | --------| -------------| -------| -------| -------| -------| -------|
> |SPIN|53.49|53.84|53.92|53.71|53.89|53.98|
> |T-SPIN|53.49|53.68|54.17|55.04|54.97|55.03|
>
> From the results, we have the following two observations:
>
> * First, both methods yield less performance enhancements on Qwen2.5-7B compared to Zephyr‑7B and Mistral‑7B. This may be attributed to: (i) Qwen2.5 is already a strong LLM, making it harder for further improvements via self-play fine-tuning; and (ii) our experiments are conducted with limited annotated samples, where achieving performance enhancements on strong LLMs is inherently challenging;
>
> * Second, SPIN still suffers from unstable optimization during iterations, which prevents further performance improvements. In contrast, T-SPIN introduces historical advantages to guide the optimization, resulting in stable performance enhancements.
>
> ---
>
> **Q2**: The extra computational overhead of generating proto-synthetic responses is not measured.
>
> **A2**: Thanks for the thorough consideration. We report the runtime, FLOPs, and memory cost for generating proto-synthetic samples below:
>
> |running-time|total flops|peak memory|
> | --------------| -------------| -------------|
> |0.8 hours|$1.8 * 10^{17}$​  FLOPs|33.43 GB|
>
> Additionally, we highlight that the cost of generating proto-synthetic samples $y_0$ is affordable for self-play fine-tuning because: (i) the generation for $y_0$ is performed **offline** and only **once** over the multi-iteration training process (see **Algorithm 1**); (ii) in each iteration, the training process takes approximately **3.5 hours**, which is significantly longer than generation, and takes up most of the total computation time.
>
> ---
>
> **Q3**: The improvement over SPIN is not significant on some datasets.
>
> **A3**: We would like to emphasize that: (i) we use only a small amount of high-quality annotated data from Ultrachat200K for training, which makes it inherently difficult to achieve significant improvements across all benchmarks; (ii) we mainly focus on overall performance over multiple benchmarks, which provides a more comprehensive view of T-SPIN’s advantages than results on individual datasets. In our experiments, T-SPIN has demonstrated higher average scores than baselines (see **Table 1**).
>
> ---
>
> **Q4**: Not compared with other improved methods based on SPIN.
>
> **A4**: Thank you for the kind reminder. Recently, we have found several studies that propose improved variants of SPIN [1, 2], and we will include comparisons with these advances in the revised version.
>
> ---
>
> **Q5**: How many iterations do you observe diminishing returns?
>
> **A5**: In experiments, we run a total of five iterations. On Zephyr, SPIN reaches its performance peak at iteration 2 and begins to decline thereafter, as shown in Figure 2(c). On Mistral, SPIN exhibited performance fluctuations starting from iteration 1 and continuing until iteration 4, as shown in Figure 5(c).
>
> ---
>
> **Reference**
>
> [1] Alami et al. Investigating Regularization of Self-Play Language Models. Arxiv, 2024.
>
> [2] Yang et al. Dynamic Noise Preference Optimization for LLM Self-Improvement via Synthetic Data. Arxiv, 2025.

---

### Note · Authors · 2025-08-12

We sincerely appreciate great efforts and valuable time of all reviewers and ACs! In the following, we provide brief final remarks of our work, including contributions, and summary of reviews and rebuttal.

---
**Contributions**. This paper aims to resolve two issues of the existing self-play method SPIN: *unstable optimization* and *training-generation misalignment*, and proposes T-SPIN with two key designs: (i) employing **historical advantage** to stablize the iterative fine-tuning process; (ii) incorporating **entropy constraint** into the self-play framework to align the training-generation metrics. Experiments on different LLMs demonstrate the superior performance of T-SPIN over its comparators. Remarkably, compared to SFT, T-SPIN achieves better performance with only 25% samples. Moreover, we also provide detailed ablation studies to verify the effectiveness of each component in T-SPIN.

**Reviews and Rebuttal**. We thank all reviewers for their constructive comments and appreciations of our work, including the well-written presentation (Reviewers Xd9e, J2Mt), the novel and useful idea (Reviewers Xd9e, J2Mt), and comprehensive exmpeiments (Reviewers Xd9e, q2aJ, J2Mt and wsFg). The main concerns with the submitted version lie in *performances on powerful LLMs and other datasets*, *computational cost*, *the isolated effect of the reward formulation and initial policy,*  and *explanations of* *reference policy and reward function*, and we have provided point-by-point responses to them in our rebuttal.

Specifically, our responses include: (i) **conducting experiments on stronger LLMs** (Reviewer Xd9e: A1; Reviewer wsFg: A1), and **validating performance on other tasks** (Reviewer J2Mt: A4); (ii) **explaining the minimal additional costs** (Reviewer Xd9e: A2; Reviewer J2Mt: A2); (iii) **comparing with variants of T-SPIN** (Reviewer q2aJ: A2, A4); (iv) **providing** **clearer explanations on reference policy and reward function** (Reviewer wsFg: A2, A3). Overall, additional experiments show the advantages of T-SPIN, and further explanations help clarify its unique design benefits.

---
In reviews, most reviewers provide positive evaluations of this work and we believe our responses can address the raised concerns. We are also confident that T-SPIN offers a novel direction for self-play fine-tuning and will further inspire other fields for LLMs. We sincerely thank the reviewers and ACs once again for their insightful suggestions and careful consideration!

---

### Decision · Program_Chairs · 2025-09-17

**Decision:**

Accept (poster)

**Comment:**

Paper proposes a new self-play play loss which, in addition to promoting moving away from the current policy to the annotated policy (prior SPIN method), also adds an additional term which promotes moving away from the initial starting policy towards the current policy policy. Authors claim that this mitigates instability of SPIN self-play whose gradient vanishes as the model policy approaches annotated policy. Second, authors also claim that replacing KL divergence with entropy in the SPIN’s regularizer mitigates empirically observed misalignment between SPIN loss and true reward. Empirical results confirming these hypotheses are very promising.

Paper is well written and motivated by a practical problem. Several reviewers asked for more experiments, ablations and clarifications. Some of these were provided during rebuttal and improved the reviewer’s assessment. Primary short comings identified by the end of the review process was lack of deeper theoretical analysis of SPIN and T-SPIN, lack of significant improvement over SPIN and lack of comparison to many variants of SPIN available in literature.